# Attention-Level Speculation

**Jack Cai** [1 2]  **Ammar Vora** [1 2]  **Randolph Zhang** [3]  **Mark O'Connor** [2]  **Mark C. Jeffrey** [1]

## Abstract

As Large Language Models (LLMs) grow in size and context length, efficient inference strategies are essential to maintain low-latency token generation. Unfortunately, conventional tensor and data parallelism face diminishing returns when scaling across multiple devices. We propose a novel form—attention-level speculative parallelism (ALSpec)—that predicts self-attention outputs to execute subsequent operations early on separate devices. Our approach overlaps attention and non-attention computations, reducing the attention latency overhead at 128K context length by up to $5\times$ and improving end-to-end decode latency by up to $1.65\times$, all without sacrificing quality. We establish the fundamental pillars for speculative execution and provide an execution paradigm that simplifies implementation. We show that existing attention-approximation methods perform well on simple information retrieval tasks, but they fail in advanced reasoning and math. Combined with speculative execution, we can approximate up to 90% of self-attention without harming model correctness. Demonstrated on Tenstorrent's NPU devices,[1] we scale up LLM inference beyond current techniques, paving the way for faster inference in transformer models.

## 1. Introduction

State-of-the-art Large Language Models (LLMs) for token generation often rely on decoder-only transformer architectures (Touvron et al., 2023; Dubey et al., 2024; Jiang et al., 2024; Yang et al., 2024; Almazrouei et al., 2023). These models stack repetitive layers of self-attention and feed-forward operations, allowing efficient scaling across layers (Tay et al., 2022; Kim et al., 2023).

[1]University of Toronto [2]Tenstorrent [3]University of Waterloo. Correspondence to: Mark C. Jeffrey <mcj@ece.utoronto.ca>.

*Proceedings of the $42^{nd}$ International Conference on Machine Learning*, Vancouver, Canada. PMLR 267, 2025. Copyright 2025 by the author(s).

[1]Our code is at github.com/mcj-group/alspec

As model sizes and context lengths increase (Chitty-Venkata et al., 2024; Shoeybi et al., 2019; Huang et al., 2019), a single accelerator can no longer meet the memory and compute demands. *Tensor parallelism* (splitting the model weights across devices) and *data parallelism* (replicating the model weights on each device) have been widely used for large-scale LLM inference. Despite their adoption, these techniques exhibit the following problems during the causal decode phase, hurting token-generation latency:

1. Tensor parallelism suffers from communication overhead. Latency for operations like *all-gather* grows with more devices, yielding diminishing returns with device scaling.
2. Data parallelism improves throughput with large batch sizes but does not improve latency.
3. As the key-value (KV) caches grow with context length, self-attention becomes a bottleneck and increases overall latency (see Section 5.1 and Appendix D for details).

We observe in Figure 1(left) that during token generation at small context length, the majority of latency comes from the non-self-attention operations, particularly the feed-forward networks. The self-attention latency grows linearly with context length. In contrast, the non-self-attention latency stays constant. This motivates our question: *Can we overlap attention with other operations to hide the increasing cost without compromising model correctness?*

We propose *attention-level speculation (ALSpec)*, replacing some full self-attentions with cheaper approximations (e.g., windowed attention with attention sink (Xiao et al., 2024)) and verifying the approximation against the true attention in parallel. When the approximation is accurate, ALSpec commits and skips expensive calculations. This form of speculation, when successful, allows non-self-attention operations to overlap in time with the self-attention operations. We find that *up to 90%* of the self-attention operations can be safely speculated without harming output quality. In contrast to decode-level speculation (Leviathan et al., 2023), ALSpec can be applied to emerging model architectures with recurrent hidden states, such as layer-looping (Eyuboglu et al., 2024) and continuous chain-of-thoughts (Hao et al., 2024). When computing resources are abundant, ALSpec could be coupled with tensor parallelism and decode-level speculation to deliver even lower latency. Figure 1 shows that ALSpec delivers up to $5\times$ reduction in

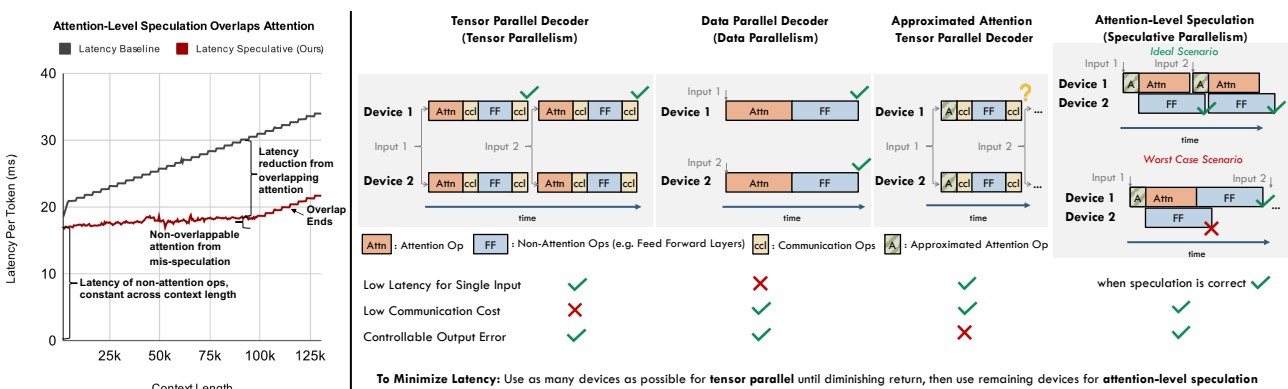

Figure 1. (*Left*) Our key result: Llama 3.1 8B decode-stage latency across context lengths on 8 devices, comparing the 8-device full tensor-parallel baseline with our 4-device tensor parallel plus 4-device speculative parallel at 87.5% speculation hit rate (see Section 5.1). (*Right*) Comparing tensor parallel (TP), data parallel (DP), attention approximation, and ALSpec when executing 2 independent inputs on 2 devices (inputs are tensor representations of previous output tokens). TP reduces per-input latency but suffers communication overheads. DP gives highest throughput but also high per-input latency. Approximation can suffer from output errors, despite latency improvements. ALSpec keeps correctness and reduces latency relative to TP when speculations are correct, with latency no worse than DP.

attention latency overhead for long-context decode at 128K. It also contrasts ALSpec with tensor-parallel, data-parallel, and approximated attention, which we detail in Section 2.

We further contribute a *speculative flash decode* kernel that computes approximate and exact attention in one pass, adding little to no overhead for speculation. Our new execution paradigm, *static graph, dynamic concurrency (SGDC) with priority gating*, maintains an op-by-op host view, while running approximate and exact attention paths concurrently. This mechanism delivers up to 1.65× speedup with 2× devices at long context lengths, even in cases where tensor parallelism fails to scale.

Our insight builds on prior observations that sparse attentions are effective (Kitaev et al., 2020; Beltagy et al., 2020; Xiao et al., 2024), but static approximate methods often fail on advanced reasoning or math tasks. By only accepting certain approximations *at run time*, we preserve accuracy across a broad set of benchmarks including reasoning, math, and information retrieval (Figure 2).

Inspired by instruction-level value prediction in CPUs, AL-Spec exploits attention sparsity in a dynamic and robust manner. Specifically, we demonstrate that a subset of challenges in transformer—safely exploiting the sparsity of self-attention—can be effectively addressed through dynamic execution with speculation. In addition to accelerating transformer-based models, it points toward new opportunities for applying generic speculation paradigms to deep learning inference, unlocking new avenues for scaling.

## 2. Motivation

Self-attention tends to dominate the latency of LLM inference at larger context lengths, as every new token attends

to an increasingly large key-value (KV) cache. Various parallelization strategies use more computing resources for lower latency or higher throughput, including *tensor parallelism* (Shoeybi et al., 2019), *data parallelism* (Huang et al., 2019), and *pipeline parallelism* (DeepSeek-AI et al., 2024). Figure 1(*right*) qualitatively compares these methods when running two inputs on two devices. Tensor parallelism splits the model weights across multiple devices to accelerate attention and feed-forward layers. Although it reduces latency for a single input, it suffers from communication overhead (e.g., *all-gather* of partial results), which grows as the number of devices increases. This yields diminishing returns when scaling past a handful of devices. Data parallelism replicates the model weights on all devices and processes multiple inputs (or tokens) simultaneously for higher throughput. However, with interactive LLM inference, the latency per input is paramount; data parallelism provides minimal improvement because each input runs on one device. Pipeline parallelism (not shown) partitions the model layers so that each device is responsible for one portion of the forward pass. Once the pipeline fills, it can achieve the same overall throughput as data parallelism. However, each input still traverses all pipeline stages, yielding a similar latency profile to data parallelism. Hence, pipeline parallelism, like data parallelism, does not fundamentally reduce inter-token decode latency.

Orthogonal to scaling up device count, *sub-quadratic algorithms* (Gu & Dao, 2024; Katharopoulos et al., 2020; Poli et al., 2023) and sparsity-exploiting *attention approximation* methods cut attention computation and memory demands. Reformer (Kitaev et al., 2020) replaces the standard dot-product attention with locality-sensitive hashing (LSH) to reduce the per-token computational complexity from linear to logarithmic in context length. SCFA (Pagliardini et al.,

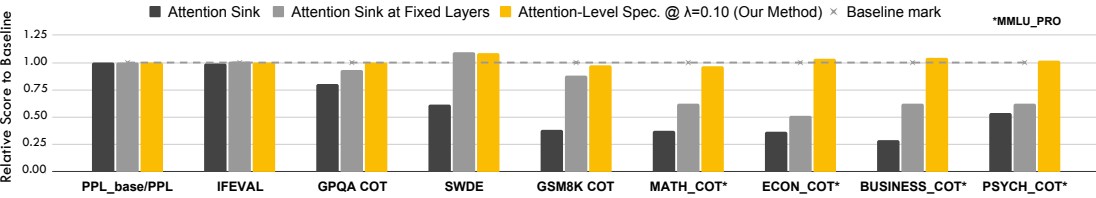

Figure 2. Correctness evaluation of attention approximation methods on various benchmarks using Llama 3.1 8B. Baseline is the model with unmodified attention. Attention sink fails for tasks requiring math and reasoning on specific subject area, such as MMLU_PRO with chain-of-thoughts (COT). Statically allowing attention sink only at specific layers reduces degradation, but still causes a noticeable gap. Dynamically using attention sink with ALSpec gives consistent correctness to the baseline for all tasks at acceptance threshold $\lambda = 0.10$.

2023) further reduces the computational cost of Reformer. Longformer (Beltagy et al., 2020) relies on sliding-window (local) attention plus a small number of global tokens. These approaches typically exhibit *static* sparsity in the sense that the approximation pattern is fixed prior to execution. Although effective in some cases (e.g., perplexity on large-context text) or with retraining, the static nature of these methods means the model execution graph is fixed prior to execution, inevitably causing quality loss in scenarios where the model needs full attention for correct results (Figure 1).

Among the approximation techniques, attention sink (Xiao et al., 2024) is simple yet surprisingly powerful. It focuses the self-attention on two sets of tokens: the first few tokens (e.g., 4) and a rolling window of $S$ recent tokens from the KV cache. This makes the effective context length of attention drastically smaller than the true context length. Attention sink achieves good performance on certain tasks (e.g., perplexity and information retrieval), even with a small $S$. However, any global information beyond the $S$-sized window or the first few tokens is inaccessible, causing attention sink to fail on tasks that require distant context or advanced reasoning. Figure 2 illustrates how attention sink on Llama 3.1 8B struggles on tasks like advanced math or reasoning (e.g., MMLU_PRO with chain-of-thought), where it yields a noticeable quality drop compared to the baseline (see Section 5.1 for methodology).

The success or failure of approximation appears to be *layer-specific*: some layers can tolerate an approximate self-attention output, while others require exact attention. Prior work shows that certain layers can be pruned or skipped without significantly impacting model quality (Elbayad et al., 2020). Consequently, one might wonder if we can *statically* choose a subset of layers to apply attention sink. To concretize this hypothesis, we design a synthetic *Needles in a Haystack* test on Llama 3.1 8B. The model inputs a 16K-token context containing 10 randomly inserted "secret keys." It must retrieve all keys from the entire text. Figure 3a shows that a model with pure attention sink ($S = 128$) recovers *none* of the secret keys. In contrast, the baseline model finds 8 of 10. Between these configurations, we create models that

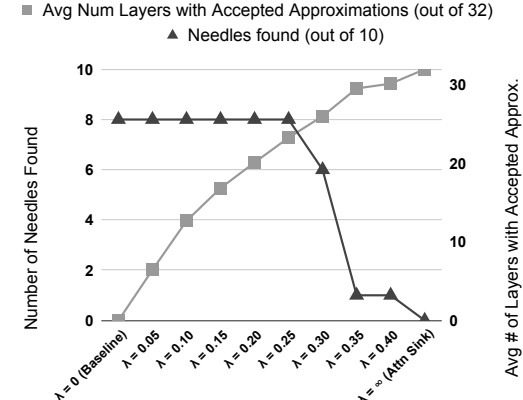

(a) Using attention sink at select layers does not harm correctness

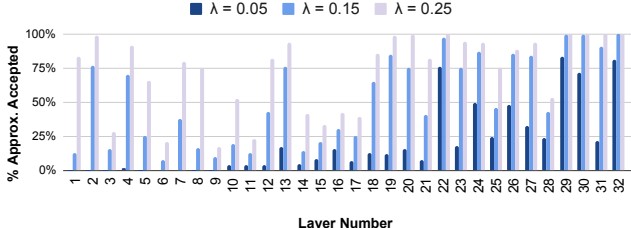

(b) Average percentage of approximation accepted per layer

Figure 3. Finding needles in a haystack with Llama 3.1 8B.

selectively use attention sink with criteria that ensure only good approximations are used: a layer accepts attention sink if, for the given token, the $L_2$ distance between the attention sink output $\tilde{A}$ and full attention output $A$ is within $\lambda \cdot L_2\_norm(A)$. A full attention sink model corresponds to $\lambda = \infty$ and the baseline model corresponds to $\lambda = 0$. Figure 3a shows that for $\lambda < 0.25$, up to $2/3$ of the layers can be approximated with attention sink without sacrificing the score. Figure 3b further shows that approximations are layer-specific: certain layers, especially the final layers, are more likely to accept approximation without harm, corroborating prior observations that deeper layers can be pruned with limited effect (Elhoushi et al., 2024).

Figure 2 shows that purely relying on a fixed layer-based approximation pattern—although better than applying atten-

tion sink to *all* layers—exhibits noticeable quality degradation when extended to a variety of tasks. For the middle bars, we create fixed layer-based approximated models, where attention sink is always used at a layer iff the likelihood to approximate that layer is greater than $p = 0.8$ in Figure 3b. The static approximation models do not balance performance gains with accuracy. Some inherently harder inputs or tasks require additional global context; even for the same task, the best approximation policy may vary from token to token or layer to layer. We therefore conclude that *no single static approximation* suffices; the key is to make decisions *dynamically* at run time.

We need a simple yet powerful way to realize dynamic execution (Barad et al., 2024). *Speculative parallelism* involves a guess about future outcomes, a verification of that guess, and error recovery when the guess is wrong. By speculating, systems can proceed optimistically and exploit opportunities for parallelism, even in the face of run-time uncertainty. Speculation is a long-standing technique in computer architecture, spanning branch, dependence, and value prediction (Rau & Fisher, 1993; Lipasti & Shen, 1996; Sazeides & Smith, 1997) to task-level speculation (Sohi et al., 1995; Herlihy & Moss, 1993; Jeffrey et al., 2015). Speculation enables concurrent or parallel execution of instructions or blocks of code, but discards results when the guesses fail. In LLMs, speculation thus far has appeared only at the decode (or token) level via speculative decoding (Leviathan et al., 2023; Hooper et al., 2023; Spector & Re, 2023; Li et al., 2024; Miao et al., 2024) which drafts several tokens in parallel. This includes decode-level speculation that employs attention sink (Sun et al., 2024).

## 3. Attention-Level Speculation Overview

*Attention-level speculation* (ALSpec) speculates on the self-attention output within each transformer layer. Specifically, ALSpec approximates the self-attention with a technique like attention sink and subsequently computes the exact self-attention. Once the exact result arrives, ALSpec *verifies* how close the approximate output is. If sufficiently close, ALSpec accepts it; if not, it reverts to the exact output. The "ideal scenario" of Figure 1 shows that speculation can overlap attention with feed-forward layers, greatly reducing latency when the prediction is correct. The "worst case" matches data parallelism latency; if the speculation fails, the computation must fall back to the exact path. This approach harnesses parallel devices (or threads) to overlap computation, hiding some of the attention latency while preventing quality degradation when approximation fails. We show that a simple approximation—namely attention sink—serves effectively as the "predictor." For each layer, we apply it to self-attention, generate a speculated attention output $\tilde{A}_i$ in layer $i$, and concurrently compute the ground-truth output $A_i$. Since attention sink can fail for certain

tasks, we design a *verification* step to ensure correctness. In practice, we find that over half of the layers can skip full self-attention by verifying that $\tilde{A}_i \approx A_i$. Meanwhile, ALSpec maintains baseline accuracy across tasks (Figure 2).

### 3.1. Low-Cost Predictor

We use a coarser version of attention sink that inputs the first and last $S$ tokens from the KV cache. We choose $S$ to be small relative to the context length (e.g., $S \in \{128, 256, 512\}$ for a $128K$ context length). In Section 4.1, we choose $S$ as the chunk size of speculative flash decode.

### 3.2. Low-Cost Verification

**Relaxed Verification via Lipschitz Continuity:** Exact equality between $\tilde{A}_i$ and $A_i$ is not mandatory to preserve correctness in LLMs; we only require that their difference remains bounded so that final outputs have small deviations. Lipschitz continuity (Scaman & Virmaux, 2018) provides a theoretical tool to relate small perturbations in one layer's output to eventual changes in final logits. Although self-attention alone is not Lipschitz continuous (Kim et al., 2021; Castin et al., 2024; Geshkovski et al., 2023), feed-forward networks, layer norms, and residual connections possess (or can be bounded by) Lipschitz constants. In addition, Geshkovski et al. show that self-attention's Lipschitz constant is bounded by $CR^2 e^{CR^2}$, where $R$ is the input magnitude bound. Combining these, we derive an upper bound on the final output's deviation in terms of per-layer differences $\|\tilde{A}_i - A_i\|$. Concretely, if each approximate attention output is within $\delta_i$ in $\ell_p$-norm of the ground-truth $A_i$, the overall output deviation, $\epsilon$, is bounded by

$$\epsilon \le \Sigma_{i=1}^{N}(1+\alpha)^{N-i+1}(1+f(R)\beta)^{N-i}\delta_i \qquad (1)$$

where $\alpha$, $\beta$, and $f(R)$ are bounded constants for the Lipschitz behavior of residuals, feed-forward networks, and self-attention inputs, respectively. A detailed proof and discussion appear in Appendix A.

**Verification Algorithm:** Motivated by the above bounds we adopt a simple threshold-based metric to accept or reject $\tilde{A}_i$. Concretely, we measure $d(\tilde{A}_i, A_i)$ in the $\ell_2$ norm and accept if: $\|\tilde{A}_i - A_i\|_2 < \lambda \cdot \|A_i\|_2$, where $\lambda$ is a small scalar hyperparameter. Algorithm 1 sketches the speculative execution procedure. For each layer, the algorithm *(i)* computes $\tilde{A}_i$ using attention sink, *(ii)* launches a new thread to continue the "post-attention" operations, *(iii)* computes $A_i$ in the main thread, then *(iv)* verifies if $\tilde{A}_i$ is sufficiently close. If yes, it commits $\tilde{A}_i$; otherwise, it discards the new thread and re-runs the next operations on the main thread's $A_i$. A simple thresholding strategy effectively ensures the algorithm does not propagate large attention errors. In addition, we show a high probability error bound on $\epsilon$ in the

**Algorithm 1** Speculative Execution with Attention Sink

> **Input:** activation $x$, N layers, threshold $\lambda$
> Start execution as the $main\_thread$.
> **for** $i = 1$ **to** $N$ **do**
>   $x = \text{ops\_before\_self\_attn}(x)$ // e.g. LayerNorm
>   $x_{spec} = \text{attention\_sink}(x)$
>   $new\_thread = \text{Thread(ops\_after\_self\_attn, } x_{spec})$
>   $x = \text{regular\_attention}(x)$
>   **if** $L_2\_\text{distance}(x, x_{spec}) < \lambda \cdot L_2\_\text{norm}(x)$ **then**
>     $main\_thread = new\_thread$
>   **else**
>     Kill $new\_thread$
>     $x = \text{ops\_after\_self\_attn}(x)$ // e.g. Feed Forward
>   **end if**
> **end for**

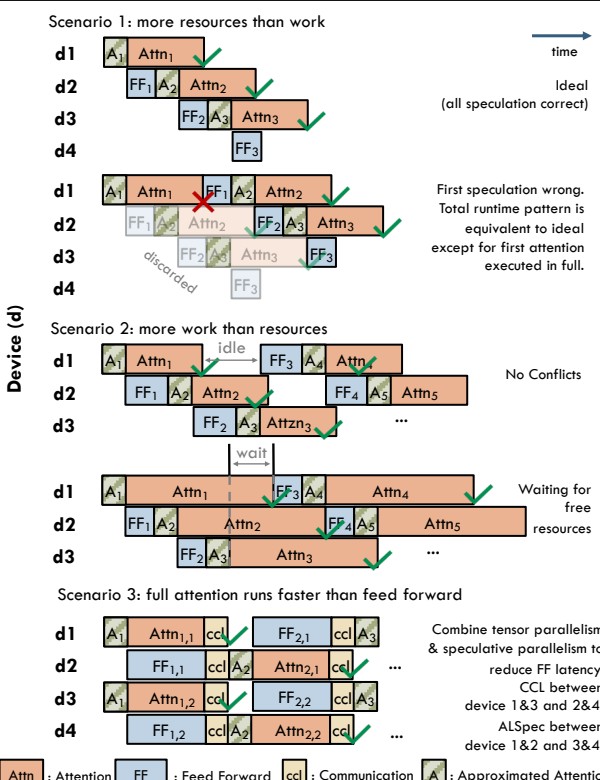

Figure 4. Algorithm 1 run-time patterns. (*Scenario 1*) *More resources than work:* Each approximate and exact path can run concurrently on separate devices. If all speculations are correct, total latency is minimized; a mis-speculation simply triggers the exact attention for one layer. (*Scenario 2*) *More work than resources:* Some computations must wait for free devices, adding stalls. Otherwise, the overlap pattern is similar to Scenario 1. (*Scenario 3*) *Feed-forward dominates:* When FF is slower than attention, we can pipeline or combine tensor parallelism with speculative parallelism (mapping devices 1:1) to further reduce FF latency.

order of $\sqrt{N}$ (Appendix A.3). This is consistent with our experimental results on deeper models (Section 5.2), achieving similar results as shallower models across benchmarks.

### 3.3. Example Speculative Executions

Figure 4 shows three high-level scenarios that can arise when applying Algorithm 1 on multiple devices in parallel. *Scenario 1: More resources than work.* With sufficiently many devices, every layer's approximate and exact attention can run in parallel without contention. In the *ideal* case, all speculations are correct, so each device can immediately continue with post-attention operations (e.g., feed-forward (*FF*) layers), yielding maximum speedup by overlapping all FF within attention. In the *mis-speculation* case, when the first approximation fails, the system reverts to full attention for that step. Subsequent layers resume the normal speculative pattern. The total latency impact of one mis-speculation remains bounded by the time needed to execute a single attention pass in full. *Scenario 2: More work than resources.* In practice, we often have fewer devices than are needed to run every speculation fully in parallel. A device may need to *wait* until a prior layer finishes using the hardware before starting its own speculation. *Scenario 3: Full attention runs faster than feed-forward.* In this scenario, speculation only uses 2 devices as the prior device will be free when FF finishes. In such cases, we combine tensor parallelism with speculative parallelism: Figure 4 illustrates a 4-device system using 2 devices for tensor parallelism and 2 devices for speculative parallelism with 1:1 mapping.

In most real-world LLM usage, FF dominates run time while attention latency grows with context length (Dao, 2024). For context lengths up to 128K tokens, FF remains the primary bottleneck, leading to *Scenario 3*, whereas *Scenario 1* or *Scenario 2* only become relevant at extremely large context lengths. Moreover, *Scenario 1 & 2* require fully pipelining or even speculating on still-speculative data (Jeffrey et al.,

2018), complicating a static execution graph. In contrast, *Scenario 3* naturally fits an *op-by-op* framework: one device is used for an approximate path and another for an exact path. No further pipelining is required. Hence, in the remainder of this paper, we focus on implementing *Scenario 3* and extending it with tensor parallelism in real systems.

### 3.4. Overheads

On the communication side, combining ALSpec and TP (*Scenario 3*) for $n$ devices reduces latency vs. full TP on $n$ devices. In each TP operation, the bandwidth-optimal all-reduce algorithm (Thakur et al., 2005) has each device send and receive $\frac{n-1}{n}$ of the tensor per all-reduce. While the total bandwidth demands per device remain nearly constant as TP device count increases, the latency of all-reduce operations during decode are often exposed on the critical path due to small activation sizes (Narayanan et al., 2021). This latency is highly dependent on the underlying network topology. For

example, in a ring topology (Sergeev & Balso, 2018), all-reduce requires $n - 1$ hops of exposed latency. In contrast, ALSpec+TP on $n$ devices performs all-reduce across $\frac{n}{2}$ devices, with $\frac{n}{2} - 1$ hops of exposed latency and adds 1 hop of exposed latency for the approximating devices to send speculated activations to the speculative FF path.

On the computation side, the ALSpec verification step introduces two $L_2$ norm calculations of the ground truth and speculated activations. This $L_2$ norm cost is similar to LayerNorm. For decode with small activations, the $L_2$ norm computation is negligible compared to FF and attention.

### 3.5. Summary

Speculative parallelism offers a simple, *dynamic* extension to static approximation. Rather than retraining or modifying model architecture, we only overlay a prediction–verification mechanism. Speculative parallelism extends naturally to other sparse attention approaches, creating a versatile framework for dynamic execution. This dynamic viewpoint transforms approximate-attention methods like attention sink from "always approximate" to "approximate only when safe," mitigating their inherent latency–quality tradeoff. By combining run-time verification with parallel computation, we overlap approximate and full-attention paths to reduce the critical path latency.

## 4. Implementation

Implementing *Scenario 3* poses significant challenges. Traditional frameworks assume an *op-by-op* execution pattern over a static dataflow graph that maps onto well-known scaling paradigms like tensor and data parallelism. By contrast, attention-level speculation requires dynamic conditional execution over the graph. An approximate path runs concurrently with computing the exact attention, and only upon verification do we commit to one.

To make the implementation feasible, we introduce two key optimizations. First, we design a custom *Speculative Flash Decode kernel* that fully hides the cost of the approximate attention path to incur minimal penalty if a speculation is rejected. Second, we introduce *static graph, dynamic concurrency (SGDC) with priority gating*: a mechanism to preserve op-by-op host code and a static computation graph, while dynamically forking and merging the speculative path under the hood. Verification then determines which result to commit, controlled by a lightweight priority gating mechanism.

### 4.1. Speculative Flash Decode

In Algorithm 1, the attention sink runs before the regular attention. In the worst case, where all speculations are rejected, the latency would be worse than pipelining. To

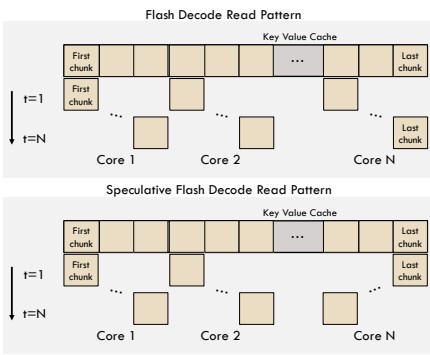

*Figure 5.* Flash decode and speculative flash decode KV cache read patterns. Speculative flash decode reads the first and last chunks first and computes a partial result as speculation.

mitigate this, we modify the read pattern of the flash decode kernel and implement an all-fused kernel that speculates for *free* while computing full attention and verification at once. Appendix B shows the full algorithm.

In short, the intuition of speculative flash decode is shown in Figure 5. The original flash decode algorithm (Dao, 2024; Dao et al., 2023) processes self-attention by chunks of KV cache from the start to the decoding position, where groups of chunks are distributed across the available cores. The results are then aggregated using the statistics. *We exploit the order invariance in which the chunks are processed.* In speculative flash decode, we set the chunk size to be $S$, assign the first and last chunk to be read first, and send the aggregated partial result immediately to the next device to execute the speculated path. The partial result is equivalently the attention sink on the first and last $S$ tokens, and can be used later towards calculating the full self-attention result. We then combine the verification step into the same single kernel, using it as a drop-in replacement for PyTorch's `scaled_dot_product_attention`.

### 4.2. SGDC with Priority Gating

Algorithm 2 implements *Scenario 3* of Algorithm 1 on two groups of devices by using the speculative flash decode kernel and a small change to the runtime, namely SGDC with Priority Gating. It uses the following methods:

- **skip_compute($p[k] == 0$):** Skips a subsequent operation's execution when $p[k] == 0$, where $k$ indicates the device index of itself.
- **all_gather(p):** Gathers $\mathbf{p}$ across $d_1, d_2$ ($d$ with the highest priority holds the correct $x$).
- **speculative_flash_decode($x$, p):** Executes a speculative decoding operation and determines the sender and receiver based on $\mathbf{p}$. Sender sets $p_k = 2$ if verification fails and otherwise $p_k = 0$. Receiver sets $p_k = 1$.

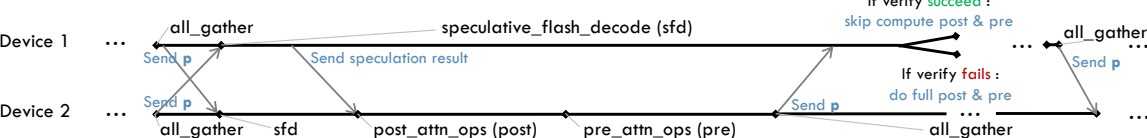

*Figure 6.* Timing diagram of speculative execution on two devices, based on Algorithm 2. We show a segment of execution timeline from the all-gather op in the first layer to the all-gather op in the second layer. When speculation succeeds, Device 1 skips the computation in post- and pre-attention ops. When speculation fails, it executes them fully.

---

**Algorithm 2** SGDC with Priority Gating

---

**Input:** replicated activation $x$, N layers, threshold $\lambda$, replicated priority vector $p$, device $d_1$, $d_2$, device id $k$. Initialize $\mathbf{p} = [p_1 = 2, p_2 = 0]$, indicating $d_1$ holds the correct $x$ with higher priority.

**for** $i = 1$ **to** $N$ **do**

    with skip_compute($\mathbf{p}[k] == 0$):

        $x = \text{ops\_before\_self\_attn}(x)$

    $\mathbf{p} = \text{all\_gather}(\mathbf{p})$

    $x = \text{speculative\_flash\_decode}(x, \mathbf{p}, \lambda)$

    with skip_compute($\mathbf{p}[k] == 0$):

        $x = \text{ops\_after\_self\_attn}(x)$

**end for**

return $x$ from the correct device based on $\mathbf{p}$

---

Figure 6 illustrates Algorithm 2. We maintain static host code where model and input activations are replicated across two devices, effectively data parallelism. Each device maintains a priority vector to track which of them holds the correct activations. Initially, both devices hold the correct activations; once speculations and computation skipping happen, only one device holds it. We perform an all-gather of the priority vector before the speculation point to determine which device has highest priority. At the speculation point, the sender device with highest priority (indicating correct activations) will execute the compute side of the speculative flash decode op, and the receiver device waits for the speculative result, then continues subsequent ops. When the full self-attention finishes, the sender will either skip the computation of the subsequent ops if the verification succeeds, or proceed executing the ops normally if the verification fails. We maintain the same ops across devices and the host dispatches all the ops asynchronously. This method can be extended to tensor + speculative parallelism: when the first group comprises $> 1$ tensor-parallel devices, the second group has the same number of devices with 1-1 mapping.[2]

---

[2]Appendix E validates SGDC with a micro-benchmark. In our real implementation, we also sync the residual stream at attention to ensure the KV cache and residual are calculated correctly. We overlap it with compute so that it does not cause extra overhead.

## 5. Experiments

We evaluate the implementation of Algorithm 2 with the speculative flash decode kernel to answer the following questions: *(i)* Does dynamic execution really improve performance on real device implementation? *(ii)* What rate of speculation hit rate do we see across benchmarks? *(iii)* Why is dynamic sometimes superior to static approximation?

### 5.1. Methodology

**Model & Hardware:** We answer the previous questions with a case study of parallelizing Llama 3.1 8B model onto 8 Tenstorrent N150 chips. The model architecture and weights are taken from the official Llama codebase without modification and implemented on Tenstorrent N150 chips, using the Tenstorrent Metalium (TT-Metalium) kernel library.[3] The model is executed with default mixed precision configuration,[4] with BF16 activations, BF8 KV cache, and BF{16,8,4} model weights. For correctness evaluations, we run the model in BF16 precision using an NVIDIA A100 or H100 GPUs; we simulate speculation by running both full and approximated attention ($S = 128$) and choosing the approximated attention output if verification succeeds.

**Benchmarks:** We evaluate the correctness and speculation hit rate measurement at various choices of $\lambda$. We group the evaluations into categories of {Question Answering (QA), Information Retrieval (IR), Reasoning, Long Context, and Math}. We use the LM Evaluation Harness (Gao et al., 2024), a framework for few-shot language model evaluation, with the default prompts and few-shot templates. As we focus on how ALSpec outputs diverge from the baseline, we do not explicitly optimize for baseline model scores.

---

[3]We conducted our experiments on Tenstorrent N150 devices using the pre-release version of the TT-Metalium v0.55.0-rc13 software stack. Tenstorrent has active ongoing software optimizations, therefore the performance results presented here are preliminary and do not reflect the latest optimized capabilities of Tenstorrent devices. Rather, they serve to illustrate the relative performance advantage provided by ALSpec. See details about TT-Metalium (Tenstorrent, 2024b), speculative flash decode, and SGDC with priority gating implementation in Appendix C, D & E.

[4]See Appendix F for TT-Metalium Llama implementation and performance measurement with tensor parallelism and ALSpec.

*Table 1.* Evaluation of attention-level speculation at various speculation verification threshold $\lambda$s for Llama 3.1 8B model across benchmarks, grouped by categories. Each entry contains (evaluated scores (0-1)) / (speculation hit rate (0-100%)). Higher $\lambda$ corresponds to more relaxed verification, allowing higher speculation hit rates, while lower $\lambda$ keeps outputs closer to the baseline.

| TASK | QA | | IR | | | MATHEMATIC | |
| | IFEVAL | GPQA COT | SWDE | FDA | GSM8K COT | MATH* | MGSM COT SW |
|---|---|---|---|---|---|---|---|
| BASELINE | 0.798 | 0.237 | 0.359 | 0.210 | 0.823 | 0.326 | 0.580 |
| $\lambda = 0.05$ | 0.799 / 69.5% | 0.257 / 38.0% | 0.371 / 57.8% | 0.205 / 37.5% | 0.806 / 49.4% | 0.332 / 21.6% | 0.592 / 53.6% |
| $\lambda = 0.10$ | 0.812 / 89.8% | 0.237 / 65.7% | 0.389 / 83.2% | 0.205 / 69.3% | 0.799 / 78.9% | 0.315 / 49.6% | 0.584 / 77.6% |
| $\lambda = 0.15$ | 0.807 / 94.8% | 0.214 / 81.1% | 0.391 / 92.9% | 0.190 / 85.7% | 0.799 / 90.2% | 0.331 / 69.3% | 0.592 / 86.0% |
| $\lambda = 0.20$ | 0.805 / 98.0% | 0.225 / 90.4% | 0.384 / 97.0% | 0.172 / 94.4% | 0.795 / 94.6% | 0.323 / 79.4% | 0.564 / 95.4% |
| $\lambda = 0.25$ | 0.794 / 99.3% | 0.208 / 95.4% | 0.359 / 98.8% | 0.158 / 98.2% | 0.719 / 96.2% | 0.266 / 86.3% | 0.516 / 96.6% |

| TASK | LONG CONTEXT | | | REASONING | |
| | HOTPOTQA | REPOBENCH-P | ECON* | BUSINESS* | PSYCH* |
|---|---|---|---|---|---|
| BASELINE | 0.920 | 0.756 | 0.328 | 0.470 | 0.289 |
| $\lambda = 0.05$ | 0.920 / 18.5% | 0.736 / 33.1% | 0.341 / 18.9% | 0.477 / 22.8% | 0.232 / 23.3% |
| $\lambda = 0.10$ | 0.915 / 42.8% | 0.768 / 59.6% | 0.339 / 45.8% | 0.490 / 50.0% | 0.296 / 52.3% |
| $\lambda = 0.15$ | 0.920 / 56.8% | 0.812 / 75.5% | 0.321 / 63.0% | 0.473 / 68.5% | 0.294 / 67.5% |
| $\lambda = 0.20$ | 0.910 / 71.6% | 0.756 / 85.0% | 0.316 / 77.5% | 0.437 / 80.7% | 0.292 / 80.9% |
| $\lambda = 0.25$ | 0.705 / 80.1% | 0.742 / 91.8% | 0.273 / 87.7% | 0.359 / 88.3% | 0.281 / 91.3% |

\* MMLU_PRO BENCHMARK WITH CHAIN OF THOUGHTS DECODING.

**Measurement:** We use the TT-Metalium profiler to measure the device kernel time of every operation during decode at specific context lengths and speculation hit rates. This performance measurement shows an ideal scalability pattern because it does not account for host and op dispatch latency, which are subject to the host configuration. We report latency per token, which is estimated by taking the measured kernel run time per layer and multiplied by the total number of layers (32). The metric token/s/user is calculated by 1/(latency per token).

### 5.2. Speculation Hit Rates and Performance Results

Table 1 shows the output quality and speculation hit rate for $\lambda \in \{0.05, 0.10, 0.15, 0.20, 0.25\}$ compared to baseline. ALSpec with $\lambda \in \{0.05, 0.10\}$ achieves on par or better correctness on all evaluated tasks, with speculation hit rates ranging from 18% (HotpotQA) to 90% (IFEval) depending on the task, and most speculation hit rates exceeding 50% for $\lambda = 0.10$. Relaxing $\lambda$ to 0.15 allows the model to speculate up to 95% of attentions successfully while doing slightly worse on GPQA and FDA. Adjusting $\lambda$ beyond 0.15 allows flexibly trading off model accuracy for computation latency, which could be helpful depending on the use case. Table 2 shows a subset of benchmarks with Llama 3 70B (80 decoder layers), confirming that a larger and deeper model achieves similar speculation hit rate with limited correctness degradation at $\lambda \in \{0.05, 0.10\}$.

Leveraging the previous experiments, we estimate performance for Llama 3 8B on N150 chips based on commonly observed ranges of the speculation hit rate: $\{50, 62.5, 75, 87.5\}$%. Our results show that ALSpec provides additional scaling when tensor parallelism scaling diminishes. Figure 7 shows that full tensor parallelism provides almost no improvements when scaling from 4 to 8 devices at long context length, whereas speculative + tensor parallelism continues

*Table 2.* Evaluation of speculative execution at various verification thresholds $\lambda$ for Llama 3.1 70B model across selected benchmarks.

| TASK | GPQA COT | GSM8K COT | HOTPOTQA |
|---|---|---|---|
| BASELINE | 0.518 | 0.958 | 0.940 |
| $\lambda = 0.05$ | 0.529 / 55.1% | 0.950 / 64.5% | 0.940 / 32.3% |
| $\lambda = 0.10$ | 0.507 / 80.0% | 0.951 / 86.6% | 0.945 / 59.0% |
| $\lambda = 0.15$ | 0.458 / 89.9% | 0.946 / 93.9% | 0.935 / 74.6% |
| $\lambda = 0.20$ | 0.446 / 95.1% | 0.936 / 96.8% | 0.935 / 83.3% |
| $\lambda = 0.25$ | 0.379 / 97.6% | 0.897 / 98.3% | 0.935 / 89.4% |

to scale. In the best case (96k context length with 87.5% speculation hit rate), ALSpec provides $1.65\times$ improvement in per-user throughput compared to full tensor parallelism.

### 5.3. Reproducibility and Estimation on GPUs

We corroborate the diminishing returns of tensor parallelism previously seen on GPUs and estimate ALSpec performance on GPUs. Chitty-Venkata et al. find that the Llama 3 8B model on A100s with DeepSpeed-MII achieves around 75, 90, and 115 tokens/s per-user throughput when scaling across 1, 2, and 4 GPUs. This scaling is similar to what we observe on N150 devices. We measure the Llama 3 8B decoding latency on 4 vs. 8 H100 GPUs doing TP using the SGLang (Zheng et al., 2024) serving framework and the FlashInfer (Ye et al., 2025) attention backend. Although we currently lack an ALSpec CUDA implementation, we estimate the ALSpec performance using SP@65% + TP4. Table 3 shows our estimation that ALSpec cuts latency on Llama 8B by $1.28\times$ vs. $1.09\times$ for full TP at context length 128K. In this case, the attention latency is only 63% of non-attention latency, suggesting greater gains for context lengths beyond 128K. Although ALSpec introduces a new attention kernel and SGDC, it does not fundamentally change the op-by-op and static graph execution style of a GPU runtime. We expect that ALSpec is implementable via CUDA graphs and conditionals (Gaiser et al., 2024).

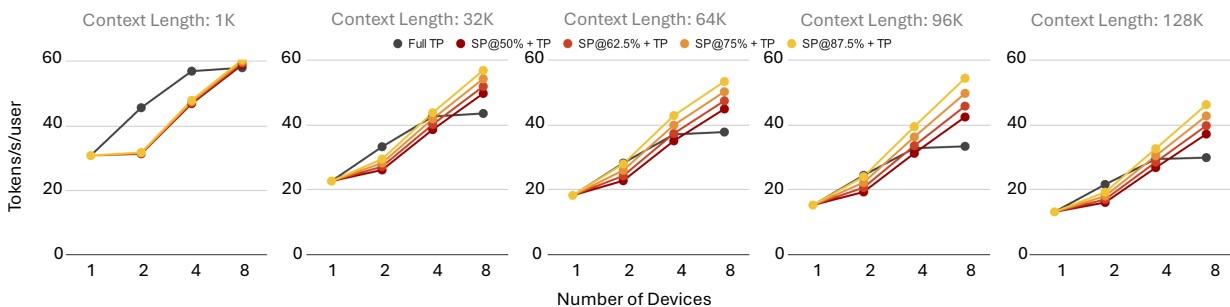

*Figure 7.* Scalability patterns of Llama 3.1 8B with full tensor parallelism (TP) and speculative parallelism at {50, 62.5, 75, 87.5}% speculation hit rate with tensor parallelism (SP@X% + TP) on Tenstorrent N150 chips. Each plot shows the scalability at a particular decode context length (1k, 32k, 64k, 96k, and 128k). Attention is speculative parallel using the speculative flash decode kernel with chunk size 256 and tensor parallel by head. Feed-forward layers are tensor parallel by inner dimension. Performance estimation is obtained by measuring the device kernel duration of every single kernel, agnostic of the dispatch and host time. At long context lengths, SP+TP (our method) continues to scale with 8 devices while full tensor parallelism scaling diminishes.

*Table 3.* Latency and throughput scaling for Llama 8B inference on H100s across various context lengths. Latencies are measured per decoder layer, and throughput is in tokens per second (tok/s). Projected ALSpec results assume 65% speculation hit rate.

| CONTEXT | 4xH100 ATTN LATENCY/LAYER (US) | 4xH100 NON-ATTN LATENCY/LAYER (US) | 4xH100 TP TOK/S | 8xH100 TP TOK/S | TP SCALING | PROJECTED ALSPEC @65% HIT RATE | ALSPEC SCALING |
|---|---|---|---|---|---|---|---|
| 1K | 13 | 95 | 244.6 | 249.3 | 1.9% | 262.0 | 7.1% |
| 32K | 29 | 100 | 214.2 | 231.3 | 8.0% | 246.0 | 14.8% |
| 64K | 49 | 100 | 191.4 | 209.7 | 9.6% | 237.8 | 24.2% |
| 96K | 56 | 100 | 178.2 | 194.2 | 9.0% | 224.9 | 26.2% |
| 128K | 63 | 101 | 169.3 | 184.8 | 9.2% | 217.5 | 28.5% |

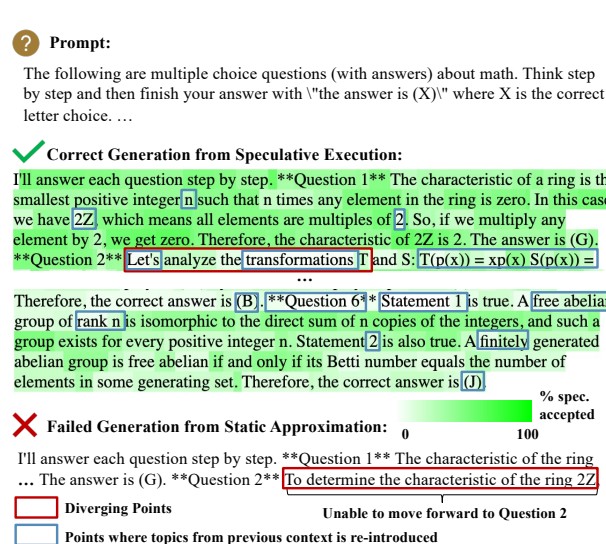

*Figure 8.* Example of output divergence in static approximation model. Divergence points corresponds to tokens with low speculation accuracy, where the static approximation model fails to adjust.

### 5.4. When Does Static Approximation Go Wrong?

In some cases, static approximation is unable to generate key tokens that influence the direction of the conversation, indicating that full attention to the context is required. This usually happens at a change of topic token. Figure 8 illustrates a specific example in MMLU_PRO_MATH where the

model is given a math problem with 5-shot examples. We visualize each token, shaded by its speculation rate, generated by the speculative execution model. The static approximation model is unable to switch to the second question during generation, hence diverging. The diverging point is a token with a low speculation rate, where it changes the topic from Question 1 to 2. Similarly, we mark the points where topics from previous contexts were introduced and they overlap with tokens with small speculation rates. Overall, dynamic execution via ALSpec detects large approximation errors and adaptively recalibrates to maintain generation quality.

## 6. Conclusion

We presented ALSpec as a new paradigm for scaling LLM inference, addressing the shortcomings of existing tensor and data parallelism methods. By leveraging predictions of attention outputs, our approach overlaps computations to reduce latency while maintaining model accuracy. The successful implementation on TT-Metalium underscores the practicality of our method. While this work focuses on speculating attention, the underlying principles of speculative parallelism are general and hold potential for wider applicability. Future research will explore the generalization of this technique, particularly in mechanisms to repair and eliminate error accumulation, as well as generic op-level speculative parallelism, which extends these principles beyond attention to optimize a broader range of ops in LLMs.

## Acknowledgments

We sincerely thank Utku Aydonat, Steven Hill, Brian Liu, Aster Plotnik, Gilead Posluns, Aditya Saigal, Angus Wu, and the anonymous reviewers for their helpful feedback. This work was supported in part by Tenstorrent through the use of N150 and GPU devices and funding from Fujitsu.

## Impact Statement

This paper presents work whose goal is to advance the field of Machine Learning. There are many potential societal consequences of our work, none which we feel must be specifically highlighted here.

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

# A. A Proof for Verification Error Bound

**Error bound** on speculation steps is essential to rigorously quantify the trade-offs between efficiency and accuracy. Such bounds ensure that the simplified model maintains acceptable performance levels while providing theoretical guarantees on the maximum potential loss in accuracy.

We accept the speculated attention output when its value is close enough to the ground truth (i.e., within a threshold $\delta$) in some metric space. To find this appropriate metric space and its corresponding $\delta$, however, requires search in an almost infinite space if done empirically.

Instead, we frame the problem as follows:

*On a model with M speculation points, given some metric d, for $\epsilon > 0$, for each speculation point $i \in 1...N$, find threshold $\delta_i$ such that for every ground truth hidden state $H_i$ and speculated hidden state $\tilde{H}_i$, if $d(H_i, \tilde{H}_i) < \delta_i$ for all i, then the distance between the final output $d(H_N, \tilde{H}_N) < \epsilon$.*

To derive the relationship between $\delta$ and $\epsilon$, we apply the Lipschitz regularity of neural network (Scaman & Virmaux, 2018). We consider the Llama (Touvron et al., 2023) architecture, consists of layer normalization, self-attention, feed-forward neural networks, and residual connections. When the self-attention output $A_i$ is being speculated as $\tilde{A}_i$, a verifier with threshold $\delta_i$ guarantees $d(A_i, \tilde{A}_i) < \delta_i$, otherwise the speculation would be rejected.

## A.1. Worst Case Error Bound

Since we are interested in bounding the error in $\mathbb{R}^n$ which is a Banach space, we denoted the distance function $d(A, B)$ as $\|A - B\|_p$ that is the p-norm of $\mathbb{R}^d$. Note that the norm induced metrics are translation invariant in Banach spaces:

$$\|X - Y\| = \|(X + A) - (Y + A)\|, \quad \forall X, Y, A \in \mathbb{R}^d$$

Those metrics also follow triangular inequality:

$$\|X - Y\| + \|X - Z\| \le \|X - Z\|, \quad \forall X, Y, Z \in \mathbb{R}^d$$

Follow the same assumption as in Section 3.2, suppose that for each layer, the self-attention output speculation error is bounded by $\|A_i - \tilde{A}_i\| < \delta_i$. Then we can easily obtain the bound for residual connections:

$$\|(H_i + A_i) - (H_i + \tilde{A}_i)\| = \|A_i - \tilde{A}_i\| < \delta_i$$

Following the residual connection of self-attention is the LayerNorm and MLP. Since both LayerNorm and MLP are Lipschitz continuous, their composite is also Lipschizt continuous.

$$\|NormMLP(A_i) - NormMLP(\tilde{A}_i))\| \le \alpha_i \|A_i - \tilde{A}_i\|$$

It follows

$$\left\|(A_i + NormMLP(A_i)) - \left(\tilde{A}_i + NormMLP(\tilde{A}_i)\right)\right\|$$
$$= \left\|(A_i - \tilde{A}_i) + \left(NormMLP(A_i)) - NormMLP(\tilde{A}_i)\right)\right\|$$
$$\le \left\|(A_i - \tilde{A}_i)\right\| + \left\|\left(NormMLP(A_i)) - NormMLP(\tilde{A}_i)\right)\right\|$$
$$< \delta_i + \alpha_i \delta_i$$
$$= (1 + \alpha_i)\delta_i$$

At the next speculation point (the self-attention in layer $i + 1$) the non-speculative and speculative inputs are $H_{i+1}$ and $\tilde{H}_{i+1}$ respectively. We denote self-attention outputs $A_{i+1}$, $\tilde{A}_{i+1}$, and $\tilde{\tilde{A}}_{i+1}$ as the non-speculative path output, speculative input followed by non-speculative output, and speculative input followed by speculative output respectively. We note that to obtain a bound for $d(A_{i+1}, \tilde{\tilde{A}}_{i+1})$, we can once again apply triangular inequality of the metric space and the result from (Castin et al., 2024), which the Lipschitz constant of self-attention, $f(R)$, is a function of the input magnitude $R$:

$$\|A_{i+1} - \tilde{\tilde{A}}_{i+1}\|$$
$$\le \|A_{i+1} - \tilde{A}_{i+1}\| + \|\tilde{A}_{i+1} - \tilde{\tilde{A}}_{i+1}\|$$
$$\le f(R)\|H_{i+1} - \tilde{H}_{i+1}\| + \delta_{i+1}$$

In general, let $H_i$ and $\tilde{H}_i$ be the non-speculative and speculative input of layer $i$, and $\alpha_i$ and $\beta_i$ be Lipschitz constants of layer $i$, then:

$$\|H_{i+1} - \tilde{H}_{i+1}\| \le (1 + \alpha_i)((1 + f(R)\beta_i)\|H_i - \tilde{H}_i\|\delta_i)$$
$$\text{where } \|H_2 - \tilde{H}_2\| \le (1 + \alpha_1)\delta_1 \tag{2}$$

Simplifying Equation by assuming the same upper bound $\alpha$ and $\beta$ across all layers, we obtain the following relationship between $\{\delta_i\}$ and $\epsilon$:

$$\epsilon = \|H_N - \tilde{H}_N\| \le \Sigma_{i=1}^{N}(1 + \alpha)^{N-i+1}(1 + f(R)\beta)^{N-i}\delta_i \tag{3}$$

## A.2. Free choice of $L_p$ and linear approximation of Equation 1

For any finite dimension Banach space, any two norms on the space are *equivalent* ($\approx$). Here we show a simplified proof for our norms $L_p$ where $1 \le p \le \infty$. Let $x \in \mathbb{R}^n$ and $1 < p < r < \infty$.

When $x = \vec{0}$, this obviously holds. Consider $x \ne \vec{0}$. Then

$$
\begin{aligned}
\|x\|_r &= (\sum_{k=1}^{n} |x_k|^r)^{1/r} \\
&\ge ((\max_{k \in n} |x_k|)^r)^{1/r} \\
&= \max_{k \in n} |x_k| \\
&= \|x\|_\infty
\end{aligned}
$$

Note that

$$
\begin{aligned}
\frac{\|x\|_r^r}{\|x\|_\infty^r} &= \frac{\sum_{k=1}^{n} |x_k|^r}{(\max_{i \in n} |x_i|)^r} \\
&= \sum_{k=1}^{n} \frac{|x_k|^r}{(\max_{i \in n} |x_i|)^r} \\
&= \sum_{k=1}^{n} (\frac{|x_k|}{(\max_{i \in n} |x_i|)})^r \\
&\le \sum_{k=1}^{n} (\frac{|x_k|}{(\max_{i \in n} |x_i|)})^p \\
&= \sum_{k=1}^{n} \frac{|x_k|^p}{(\max_{i \in n} |x_i|)^p} \\
&= \frac{\|x\|_p^p}{\|x\|_\infty^p}
\end{aligned}
$$

Which gives

$$\|x\|_r = \|x\|_\infty (\frac{\|x\|_r^r}{\|x\|_\infty^r})^{1/r} \le \|x\|_\infty (\frac{\|x\|_p^p}{\|x\|_\infty^p})^{1/r} \le \|x\|_\infty (\frac{\|x\|_p^p}{\|x\|_\infty^p})^{1/p} = \|x\|_p$$

Note this holds for $p, r > 0$. We can use the same argument to show that $\|x\|_p \le \|x\|_1$. Also,

$$
\begin{aligned}
n\|x\|_\infty &= \sum_{k=1}^{n} \max_{i \in n} |x_i| \\
&\ge \sum_{k=1}^{n} |x_k| \\
&= \|x\|_1
\end{aligned}
$$

This shows that $\|x\|_\infty \le \|x\|r \le \|x\|_p \le \|x\|_1 \le n\|x\|_\infty$. Using this fact, the choice of $l_p$ would at most differ by a constant of $n$ that only depends on the dimensionality but not the magnitude of the input vector.

When $\alpha$ and $\beta$ are small, we can approximate the error bound linearly. We have that

$$
\begin{aligned}
\epsilon &\le \Sigma_{i=1}^N (1+\alpha)^{N-i+1}(1+f(R)\beta)^{N-i}\delta_i \\
&= \sum_{i=1}^N \left( \sum_{j=0}^{N-i+1} \binom{N-i+1}{j} \alpha^{N-i+1-j} \right) \left( \sum_{j=0}^{N-i} \binom{N-i}{j}(f(R)\beta)^{N-i-j} \right) \delta_i \\
&\approx \sum_{i=1}^N \left( \sum_{j=0}^{N-i+1} \binom{N-i}{j} \alpha^{N-i-j} \right) \left( \sum_{j=0}^{N-i} \binom{N-i}{j}(f(R)\beta)^{N-i-j} \right) \delta_i \\
&\approx \sum_{i=1}^N \left( 1 + (N-i)\alpha \right) \left( 1 + (N-i)f(R)\beta \right) \delta_i \\
&\approx \sum_{i=1}^N \left( 1 + (N-i)(\alpha + f(R)\beta) \right) \delta_i \\
&\approx \left( N + \frac{1}{2}N^2 \right)(\alpha + f(R)\beta)\delta
\end{aligned}
$$

Which $\delta \ge \delta_i$ is an overall bound. This bound holds for the worst case; all errors are in the same direction. However, this is highly unlike in practice and it has a dependence on $N^2$. As a result, the threshold chosen using this bound for rejecting speculation is too small and not useful.

### A.3. High Probability Error Bound

However, with stronger assumptions, we can obtain a better error bound. Assume that the errors in the speculation rounds are independent and follow some distribution with mean 0. In other words, the direction of the error will not be biased towards any direction. By rejecting errors that exceeding some bound, we ensure that speculated results are bounded by the ball centered on the ground truth $x$, with radius $\lambda \cdot \|x\|$. This is a martingale:

$$
\mathbb{E}[\tilde{H}_{i+1} - H_{i+1}|\tilde{H}_i] = 0
$$

With the causal softmax attention

$$
H = V \text{softmax}(QK^T)
$$

The approximation is obtained as

$$
\tilde{A} = QK^T(:, B)
$$

And

$$
\tilde{H} = V(B, :)\text{softmax}(\tilde{A})
$$

Where $B$ is a subset of columns of $K^T$. With the error being bounded:

$$
\|H_{i+1} - \tilde{H}_{i+1}\|_2 \le \lambda\|\tilde{H}_i\|_2
$$

With Azuma-type inequality for Banach space-valued martingales (Naor, 2012), there exists some universal constant $c > 0$ such that.

$$
\mathbb{P}(\|\tilde{H}_N - H_N\|_2 \ge \epsilon) \le e^4 \cdot \exp\left( -\frac{c\epsilon^2}{\sum_{i=1}^N \lambda^2\|\tilde{H}_i\|_2^2} \right)
$$

Rearranging the equation, with probability $\delta \in (0, 1)$, we have

$$
\epsilon \le \sqrt{\frac{(4 - \log\delta)\lambda^2 \sum_{i=1}^N \|\tilde{H}_i\|_2^2}{c}}
$$

Choose $\delta = 0.05$, with some addition assumption that $\|H_i\|_2$ are "on the same scale" due to layer norm, we replace them with a universal constant $\|H\|_2$. Now with probability 95%,

$$\|\tilde{H}_N - H_N\|_2 \leq \sqrt{\frac{(4 - \log 0.05)\lambda^2 N \|H\|_2^2}{c}}$$

$$\frac{\|\tilde{H}_N - H_N\|_2}{\|H\|_2} \leq \lambda \sqrt{\frac{7N}{c}}$$

This bound has much better dependence on $N$ which is the total number of speculations than the last bound. However, it relies on unbiased assumption on speculation error that is yet to be proved. In practice, $\lambda = 0.1$ worked well in varies benchmarks and this bound is much closer to 0.1.

# B. Speculative Flash Decode Algorithm

The speculative flash decode algorithm builds upon the standard Flash Attention 2 (Dao, 2024). In the scope of this work, we only consider the forward pass for inference application. During the decode stage, the activation query matrix ($\mathbf{Q}$), with sequence length 1, is very small and fits into the on-chip SRAM. The keys ($\mathbf{K}$) and values ($\mathbf{V}$) matices, particularly at large context length, are large and can only live in the slower DRAM. In flash decode, input $\mathbf{Q}$ is replicated across parallel workers and each worker computes partial attention output (with local statistics) using a portion of $\mathbf{K}$ and $\mathbf{V}$ by running the standard flash attention algorithm with chunk size $S$. Next, the partial outputs are gathered and reduced on the reducer worker using the local statistics by applying the online softmax trick. In the extreme case of having only a single worker, flash decode is the same as flash attention. In the algorithm below, we skip the details of how flash attention is computed and we simply call $\texttt{O,m,l = flash\_attention(Q,K,V)}$ to obtain the partial output $\mathbf{O}$ and statistics $m$ & $l$.

Speculative flash decode builds upon the flash decode algorithm with a few modifications. The kernel consists of the sender, which computes the full and speculative attention outputs, and the receiver, which waits for the speculative results to arrive. First, one (or two) of the workers compute the partial output using first and last chunk of $\mathbf{K}$ and $\mathbf{V}$, which the result serves as the speculated attention output. Then, the worker continues to finish the rest of its own assigned $\mathbf{K}$ and $\mathbf{V}$ chunks. Finally, the reducer worker computes the full output using the partial outputs from each worker and the speculative output. We show speculative flash decode in Algorithm 3 below:

---

**Algorithm 3** Speculative Flash Decode (forward pass for inference)

---

1: **Require:**
2:      $n_{\text{workers}}$: total number of parallel workers
3:      $m$: total number of chunks
4:      $\mathbf{Q}$: query of size $1 \times d$ (fits in on-chip SRAM)
5:      $\{\mathbf{K}_1, \ldots, \mathbf{K}_m\}$: key chunks in DRAM
6:      $\{\mathbf{V}_1, \ldots, \mathbf{V}_m\}$: value chunks in DRAM
7:      Routines:
8:         $\texttt{flash\_attention}(\mathbf{Q}, [\mathbf{K}], [\mathbf{V}]) \rightarrow (\mathbf{O}, m, l)$
9:         $\texttt{send\_to\_receiver}(\cdot), \texttt{send\_to\_output\_worker}(\cdot)$
10:         $\texttt{online\_softmax\_combine}(\cdot)$

11: **// SENDER (run on each worker)**
12: $\texttt{chunks\_per\_worker} \leftarrow \lceil m/n_{\text{workers}} \rceil$
13: $\texttt{assigned\_chunks} \leftarrow$ range of chunk indices for this worker, excluding 1 and $m$
14: **if** worker_id $= 1$ **then**
15:      $(\mathbf{O}_s, m_s, l_s) \leftarrow \texttt{flash\_attention}\big(\mathbf{Q}, [\mathbf{K}_1, \mathbf{K}_m], [\mathbf{V}_1, \mathbf{V}_m]\big)$ {Speculative output}
16:      $\texttt{send\_to\_receiver}(\mathbf{O}_s)$
17:      $\texttt{send\_to\_output\_worker}(\mathbf{O}_s, m_s, l_s)$
18: **end if**
19: $(\mathbf{O}^{(i)}, m^{(i)}, l^{(i)}) \leftarrow \texttt{flash\_attention}(\mathbf{Q}, [\mathbf{K}_j \mid j \in \texttt{assigned\_chunks}], [\mathbf{V}_j \mid j \in \texttt{assigned\_chunks}])$
20: $\texttt{send\_to\_output\_worker}(\mathbf{O}^{(i)}, m^{(i)}, l^{(i)})$

21: **// OUTPUT WORKER (on sender)**
22: Gather all partial outputs $(\mathbf{O}^{(i)}, m^{(i)}, l^{(i)})$ from each worker
23: Gather speculative $(\mathbf{O}_s, m_s, l_s)$ from worker 1
24: $(\mathbf{O}, m, l) \leftarrow \texttt{online\_softmax\_combine}\Big(\{(\mathbf{O}^{(i)}, m^{(i)}, l^{(i)})\}_{i=1}^{n_{\text{workers}}} \cup \{(\mathbf{O}_s, m_s, l_s)\}\Big)$
25: **return** $\mathbf{O}$

26: **// RECEIVER**
27: **wait** for speculative output $\mathbf{O}_s$ to arrive from worker 1

---

# C. Tenstorrent N150 Chips Specification and Tenstorrent Metalium (TT-Metalium) Framework

## C.1. Tenstorrent Overview

Tenstorrent specializes in high-performance computing solutions for artificial intelligence (AI) and machine learning (ML) workloads. Their primary focus is on designing innovative hardware accelerators and software frameworks that enable efficient inference and training of deep learning models. By leveraging custom processor architectures optimized for AI workloads, Tenstorrent aims to push the boundaries of energy efficiency and computational throughput.

## C.2. N150 Wormhole Chips

The N150 Wormhole chip is an AI accelerator designed for High Performance Computing (HPC) applications such as high-throughput and low-latency machine learning inference. These chips are designed to be highly flexible and scalable. Each chip contains a mesh of 72 Tensix™ compute cores connected via 2 network-on-chip (NOCs), offering 262 Tera-FLOPs of FP8 performance. Moreover, with on-board ethernet, N150 chips can be interconnected in various topologies consisting of 2, 4, 8, and 32 N150 devices.

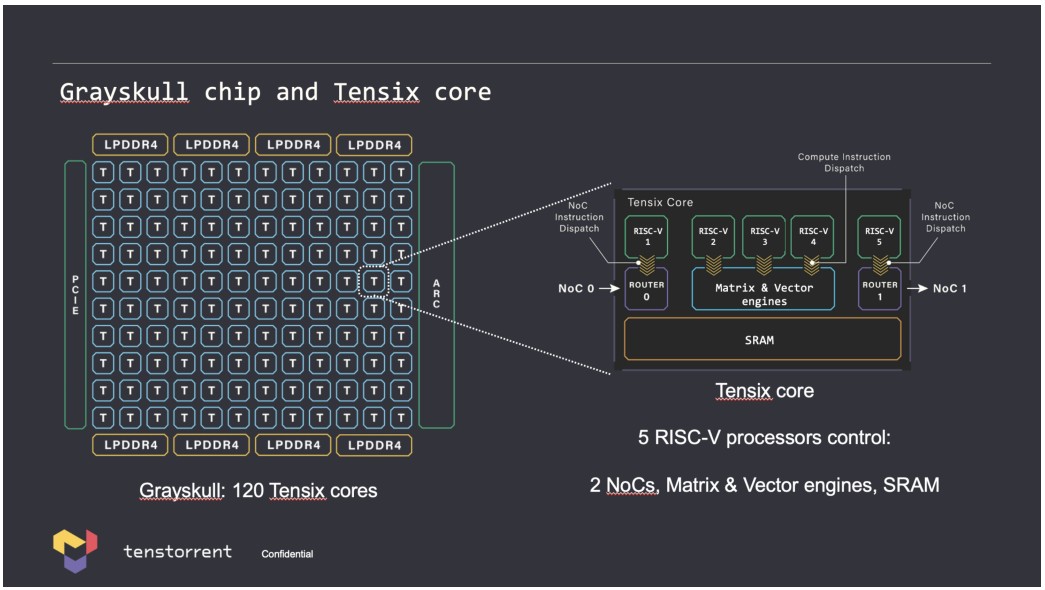

*Figure 9.* Architecture of a GraySkull™ device, the predecessor to the Wormhole™. Image taken from (Tenstorrent, 2024c).

## C.3. Tensix™ Core

The Tensix™ core is a crucial building block on the Wormhole device. It consists of 5 small RISC-V processors (also known as Baby RISCVs), that handle compute, data movement, and dispatch of instructions. For compute, a Matrix Engine (FPU) is used to perform various binary operations on small 32x32 matrices (known as tiles), such matrix-multiplication, dot-product, and other element-wise operations. Alongside, a Vector engine (SPU) is used for special operations such as GELU, exponential, and sqrt. For memory, each core has access to 1.5MB of high-bandwidth SRAM (L1) as well as 12GB of GDDR6 off-chip DRAM. A Data movement engine is also available on each core, connected to 2 NOCs. By creating a mesh of cores via these NOCs, the N150 provides access to a high capacity of 108 MB of SRAM, allowing for the Tensix™ cores to operate at "silicon peak" performance. To allow users to leverage the powerful capabilities of the N150 chip, Tenstorrent has developed a low-level kernel library: TT-Metalium (Tenstorrent, 2024c).

## C.4. TT-Metalium

TT-Metalium, also known as TT-Metalium, is an open-source low-level kernel operations library that allows users to enable high-performance machine learning applications on Tenstorrent hardware, such as the N150 chip. At a fundamental level, TT-Metalium assigns a 1:1 mapping between each Tensix™ core and a thread, discarding the need for complex thread

scheduling. Using TT-Metalium, users can leverage bare-metal programming to write kernels in C/C++ that run directly on each Tensix[TM] core and perform computation on tilized input tensors. With the fine-grained control enabled by the TT-Metalium framework, the mesh of Tensix[TM] cores on the N150 chip can be leveraged in a robust and powerful manner, overall supporting highly intensive machine learning workloads.

### C.5. Kernel programming with TT-Metalium

Programming a Tensix[TM] core on TT-Metalium mainly consists of writing three kernels – a reader, writer, and compute. There are other advanced types of kernels available, such as dispatch and ethernet data movement kernels, however, they are not relevant to the user when bringing up general operations, and can be accessed via an abstraction layer. To implement an operation, the reader kernel is programmed to read data from L1 or DRAM memory and pass it off to the compute kernel via local buffers in L1. After the compute kernel has completed the operation (using the FPU/SFU), it packs out the data into another local buffer, where it is then accessed by the writer kernel. The writer kernel is responsible for formatting the data as required and providing outputs either in L1 or DRAM. As these three kernels run in parallel, there is native support for overlapping data movement with computation, where the reader/writer kernel can transfer data while the compute kernel is busy. It is also important to note that the compute kernel only supports tilized inputs to maximize utilization, and as such, the reader/writer kernels must format input tensors in this manner. Moreover, a set of data movement APIs, that leverage the NOC, are available inside each reader/writer kernel, allowing each core to access the L1 memory of any other core. Putting all of this together, TT-Metalium can be used to create efficient data patterns across the N150 chip, resulting in powerful implementations of operations used in machine learning.

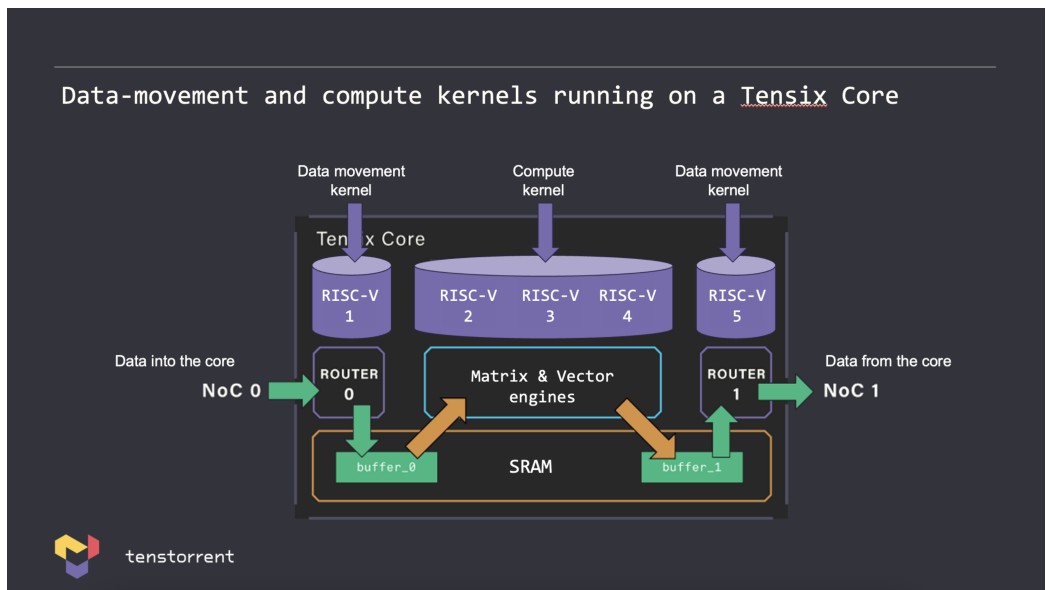

*Figure 10.* The workflow of a Tensix[TM] core. Image taken from (Tenstorrent, 2024c).

### C.6. TT-NN and Execution Paradigm

Powered by TT-Metalium in the background, TT-NN is a user-friendly library implemented in Python that enables fast bring-up of machine learning models without sacrificing performance. It provides easy-to-use APIs to perform machine learning operations on N-dimensional tensors, similarly to PyTorch. Additionally, TT-Metalium objects such as a Multi-Device tensor are exposed in TT-NN, allowing users to easily implement algorithms and paradigms that span multiple N150 chips and leverage distributed computing. When running models implemented in TT-NN, operations are dispatched by the host and scheduled to run on Tenstorrent hardware via the PCIe link. Once results are ready on device, they are sent back to the host.

## D. Speculative Flash Decode Implementation on TT-Metalium

We implement the speculative flash decode kernel on TT-Metalium based on the existing flash decode kernel implementation (Cai, 2024). All kernels on TT-Metalium are programmed directly on the Tensix$^{TM}$ cores to take advantage of the parallelism and tile-based granularity offered by the hardware. Each N150 chips consists of 64 worker cores and math is done on tiles with size $32 \times 32$. In the case of flash decode, we parallelize the batch and key/value heads across the cores, and put the query heads (usually $\leq 32$) within a tile.

We illustrate this idea using Llama 3.1 8B's group query attention, which contains 32 query heads and 8 k/v heads, with head dimension 128. Assuming batch 1, during decode, query has sequence length of 1 so the shape of **Q** is $[S = 1, B = 1, H = 32, D = 128]$, and the shape of shape of **K** and **V** are $[B = 1, H = 8, S, D = 128]$. In this case, we parallelize the k/v heads across the 64 cores, with 8 cores per head. Each of the 8 cores then each takes a chunk of **K** and **V** fractured in the sequence length dimension and run the speculative flash decode algorithm (Appendix B). We illustrate the implementation in Figure 11:

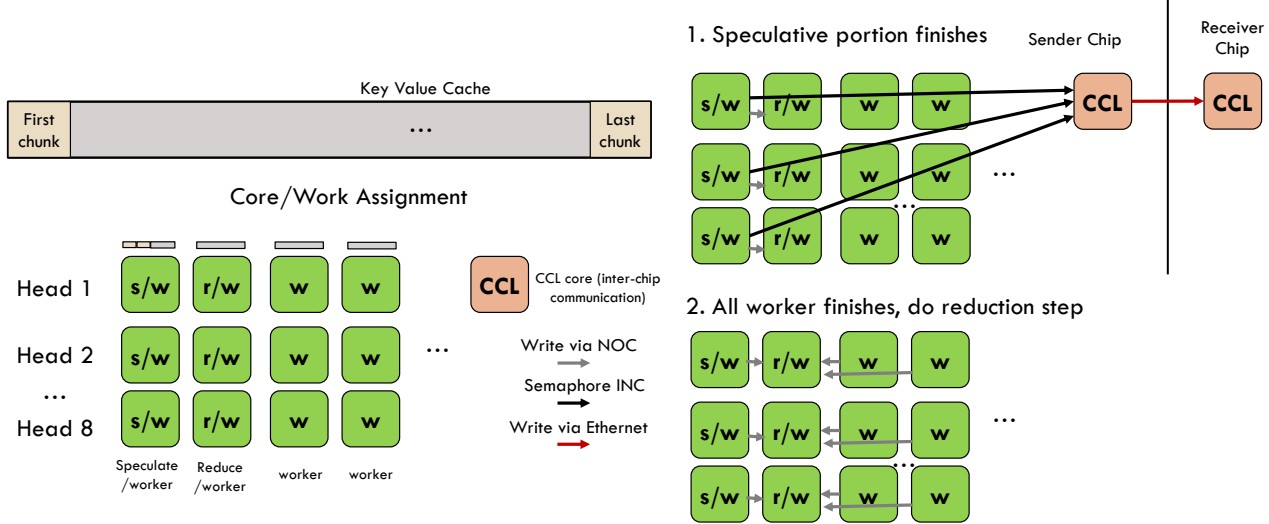

*Figure 11.* Speculative flash decode implementation on TT-Metalium. Each k/v head is split across 8 cores in the sequence length dimension. For each head, one core computes speculative and partial attention output, one core computes partial attention output and reduction, and the rest of the cores computes partial attention output only. When speculative results is finish, the worker core signals the CCL core to send result to the receiver chip via ethernet link. The rest follows the standard flash decode implementation, where partial outputs are gathered and reduced on the reduction core. In the end, we fused the verification step and update the priority vector.

Overall, speculative flash decode yields almost no overhead compared to standard flash decode in our performance measurement. The speculative portion computes towards the final output, and communication across chip is done on separate core via ethernet. When parallelizing attention across multiple devices using tensor parallelism, we split the k/v heads across the chip so that each head gets more workers per head. For example, splitting onto 2 chips gives 4 k/v heads per chip, and split across 8 chips gives 1 k/v head per chip. In practice, parallelizing across the heads yields diminishing returns when each chip receives "too little" work. In the case of having all 64 cores computing a single head, the bottleneck is NOC rather than compute due to the NOC transactions at the reduction step. For this reason, we see diminishing returns when scaling to 8 devices doing full tensor parallelism, and a better alternative would be doing 4 device tensor parallelism + ALSpec for scaling up to 8 devices. Figure 12 demonstrates speculative flash decode, sender & receiver side kernel durations, across context length for tensor parallel Llama 3.1 8B attention across 1,2,4 & 8 devices with chunk size {128, 256, 512}.

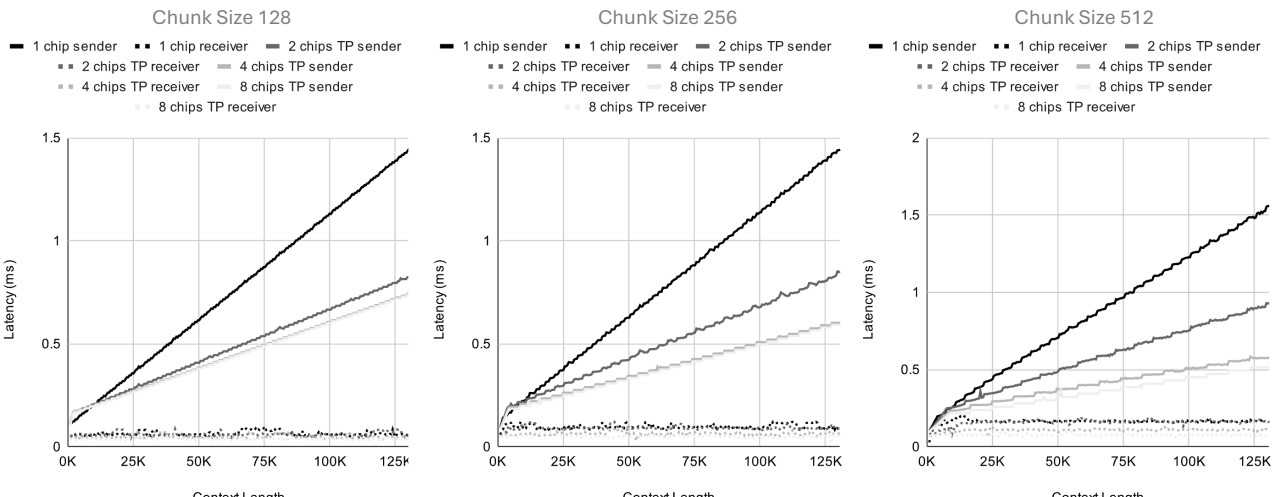

*Figure 12.* Speculative flash decode kernel duration for Llama 3.1 8B shape with chunk size {128, 256, 512}.

## E. SGDC with Priority Gating Implementation on TT-Metalium

The static graph dynamic concurrency (SGDC) paradigm, refers to having a static operation graph generated by the host but having the devices execute concurrently based on a function, which in our case is the priority gating mechanism. For the speculative flash decode algorithm, the dynamic concurrency comes in the form of % speculation acceptance and the resulting overlap between attention and post-attention operations, which varies layer-to-layer and token-to-token. To support this paradigm, we made a change to the run-time behavior of the TT-Metalium framework.

We leverage the fact that any kernel for any operation (data movement, compute, ethernet) in TT-Metalium is launched through a small set of C++ kernel source code files. By passing a flag at the build stage of these kernel source code files, we can control the behaviour of the kernel launch stage. At the start of model execution, we create a multi-device priority vector 'p', where each device tensor contains the device's priority. This tensor will have the same lifetime as the entire model and is used as the gating mechanism. Then, we pass the base memory address of this tensor to the kernel source code files using a C++ define, *SKIP_COMPUTE*. When a kernel is launched, it first passes through the kernel source code, where the address specified by *SKIP_COMPUTE* is used to check whether the kernel should be launched. As described in the paper, if the priority of the device is 0, then the operation will be skipped. Note, the ability to use the *SKIP_COMPUTE* can be toggled by adding/removing the define from the build arguments of the kernel files. We make sure to remove the define before running the *speculative_flash_decode* kernel, so as to not skip this operation.

Using this mechanism, we can enable the behaviour described by scenario 3 in Figure 5. When device 1 is a sender device and finds that the speculated result is correct within a certain threshold, it can completely skip the post-attention operations using the *SKIP_COMPUTE* flag, relying on the concurrent computation performed on device 2. When viewed from the host, this results in a static operation graph, however at the device level, we have achieved dynamic concurrency.

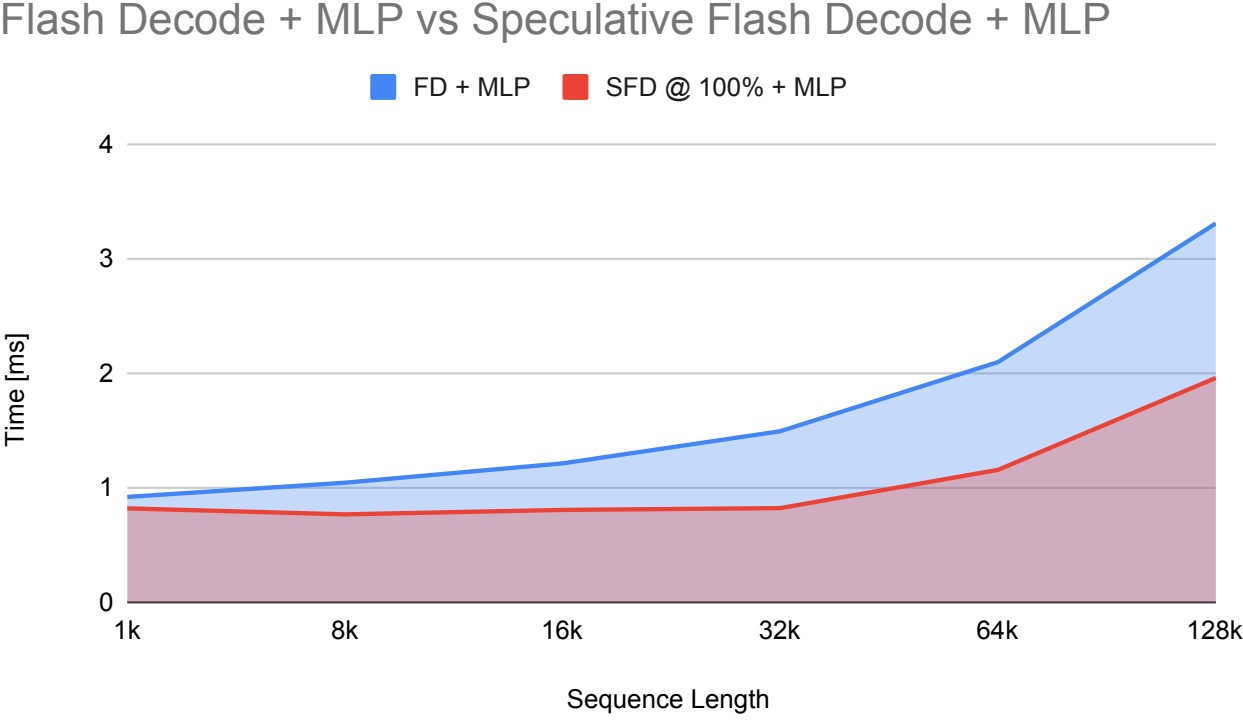

*Figure 13.* End-to-end performance using Llama3.1 8b shapes for Flash decode followed by MLP, compared to Speculative Flash Decode at 100% specualation followed by MLP. For FD one device is used and for SFD two devices are used.

## F. Llama 3.1 8B Implementation on TT-Metalium and Performance Estimation

TT-Metalium supports native mixed precision computation at 4 different data formats on N150 chips: float32, Block Floating Point (bfloat) 16, bfloat8_b, and bfloat4_b. These data formats can be used interchangeably within the model execution to opportunistically speed up computation when full precision is not required. In Llama 3.1 8B's implementation, 4/8-bit feedforward weights and 8-bit precision KV cache has been used to reduce memory bottleneck, while operations sensitive to precision, such as rotary embedding, are performed in full or half precisions. To ensure correctness, op, module, and model level unit tests are performed to ensure output quality. Specifically, Pearson Correlation Coefficient (PCC) and top 1/5 scores has been used as common metrics in unit tests. Generally, quantization is applied to models only if the PCC score is above 0.99 and top 1/5 scores are above 0.9/0.99. The empirically verified strict thresholds ensure that the model benefits as much as it can from the mixed-precision inference without hurting the output quality. Therefore, to ensure our performance estimation of ALSpec matches and benefits the existing Llama model implementations on TT-Metalium, we use the default mixed-precision models brought up on the repository (Tenstorrent, 2024a).

We estimate the pure device performance of the Llama 3.1 8B model during decode phase using the following method:

- We measure the kernel latency (Device FW Duration) of the operations within a single decoder layer, *excluding* the flash decode op. We call this per_layer_latency.

- We also measure the kernel latency of operations outside the decode (embedding and LM-head ops). We call this fixed_latency.

- We sweep the latency of flash decode and speculative flash decode op across context length, from 1K to 128K. We denote the latency $FD(K)$ and $SFD(K)$ for the latency at context length K.

- For speculative flash decode, we record both the sender ($SFD_S(K)$) and receiver ($SFD_R(K)$) kernel latency.

- We repeat the measurement on 1, 2, 4 and 8 devices tensor parallelism.

For decode phase, the input is always the same shape, but only the KV cache gets longer as the context length increase. In the Llama 3 implementation, the only operation that changes latency with respect to context length is the flash decode operation. Therefore, with 32 layers in the Llama 3.1 8B model, we can obtain the token latency and tokens/s/user throughput (1/(token latency)) at context length K using the following formula for full tensor parallelism:

$$\text{token\_latency} = \text{fixed\_latency} + 32 \cdot (\text{per\_layer\_latency} + FD(K))$$

To estimate the token latency at a particular speculation hit rate $r$ [5], we use the following formula:

$$\text{token\_latency} = \text{fixed\_latency} + \lfloor 32 \cdot r \rfloor \Big( SFD_R(K) + \max(SFD_S(K) - SFD_R(K), \text{per\_layer\_latency}) \Big)$$
$$+ \lceil 32 \cdot (1 - r) \rceil \cdot \Big( \text{per\_layer\_latency} + SFD_S(K) \Big)$$

In short, this formula takes the overlapped attention latency when speculation hits, and full layer latency when speculation misses. It serves as an upper bound for latency because we always round down for the number of layers with speculation hits [6].

We present the full latency breakdown for per_layer_latency and fixed_latency across 1, 2, 4 and 8 devices tensor parallel in Table 4, 5, 6, & 7. One can verify the estimation in Figure 7 based on the table and the speculative flash decode latency in Figure 12. The latency breakdown also shows how diminishing returns happen with tensor parallelism, while compute operations such as matmul get lower latency when parallelized to more devices, CCL operation latency such as all-gather increases, making the overall latency reduction minimal.

---

[5]Note that in our measurement, we take $r$ as a multiple of $1/32$ to ensure that full layers are doing speculation.

[6]This formula doesn't account for the last attention speculation, which should overlap with the fixed latency (LM Head). We are aware of this limitation and we use the formula for its simplicity. In the amortized case of running multiple token generation, this approximation holds as the final attention can overlap with the pre-attention ops of the next token.

*Table 4.* Llama 3.1 8B latency break down on 1 x N150 chip

| **Per Layer** | | **Embedding & Lm Head** | |
| --- | --- | --- | --- |
| OP CODE | DEVICE FW Duration [ns] | OP CODE | DEVICE FW Duration [ns] |
| LayerNorm | 12454 | Embeddings | 5745 |
| Matmul | 109174 | Embeddings | 5639 |
| ShardedToInterleaved | 4299 | Transpose | 13808 |
| NLPCreateHeadsDecode | 21048 | Transpose | 13512 |
| RotaryEmbeddingLlama | 4828 | Slice | 2091 |
| RotaryEmbeddingLlama | 4827 | Slice | 2089 |
| PagedUpdateCache | 13166 | InterleavedToSharded | 1694 |
| PagedUpdateCache | 13049 | InterleavedToSharded | 1759 |
| ScaledDotProductAttentionDecode | 69853 | Reshard | 3386 |
| InterleavedToSharded | 1701 | LayerNorm | 13243 |
| NLPConcatHeadsDecode | 3055 | Matmul | 546986 |
| Matmul | 76389 | ShardedToInterleaved | 9003 |
| Binary | 1823 | Matmul | 545955 |
| LayerNorm | 12487 | ShardedToInterleaved | 8833 |
| Matmul | 141261 | Matmul | 545707 |
| Matmul | 141772 | ShardedToInterleaved | 8944 |
| Binary | 30204 | Matmul | 547068 |
| Matmul | 252556 | ShardedToInterleaved | 9055 |
| Binary | 2654 | Concat | 29186 |
| Total / Layer without flash decode | 846747 | Total LM Head | 2313703 |

*Table 5.* Llama 3.1 8B latency break down on Tensor Parallel 2 x N150 chips.

| **Per Layer Context Length 1K** | | **Embedding & Lm Head** | |
| --- | --- | --- | --- |
| OP CODE | DEVICE FW Duration [ns] | OP CODE | DEVICE FW Duration [ns] |
| AllGather | 19567 | | |
| LayerNorm | 12471 | | |
| Matmul | 56690 | | |
| ShardedToInterleaved | 2541 | Embeddings | 5549 |
| NLPCreateHeadsDecode | 10972 | Embeddings | 5767 |
| RotaryEmbeddingLlama | 4834 | Transpose | 13570 |
| RotaryEmbeddingLlama | 4820 | Transpose | 13590 |
| PagedUpdateCache | 8447 | Slice | 2094 |
| PagedUpdateCache | 8383 | Slice | 2222 |
| ScaledDotProductAttentionDecode | 64309 | InterleavedToSharded | 1748 |
| InterleavedToSharded | 1833 | InterleavedToSharded | 1701 |
| NLPConcatHeadsDecode | 2320 | AllGather | 20607 |
| Matmul | 41730 | LayerNorm | 12931 |
| ReduceScatter | 21842 | Matmul | 546309 |
| Binary | 1621 | ShardedToInterleaved | 8779 |
| AllGather | 20887 | Matmul | 546729 |
| LayerNorm | 12486 | ShardedToInterleaved | 8992 |
| Matmul | 76942 | Concat | 17048 |
| Matmul | 77430 | | |
| Binary | 16543 | Total LM Head | 1207636 |
| Matmul | 128695 | | |
| ReduceScatter | 19108 | | |
| Reshard | 3233 | | |
| Binary | 1620 | | |
| Total / Layer without flash decode | 555015 | | |

*Table 6.* Llama 3.1 8B latency breakdown on Tensor Parallel 4 x N150 chips.

**Per Layer**

| OP CODE | DEVICE FW Duration [ns] |
| --- | --- |
| AllGather | 22378 |
| LayerNorm | 12485 |
| Matmul | 39022 |
| ShardedToInterleaved | 2057 |
| NLPCreateHeadsDecode | 6269 |
| RotaryEmbeddingLlama | 4837 |
| RotaryEmbeddingLlama | 4834 |
| PagedUpdateCache | 6260 |
| PagedUpdateCache | 6342 |
| ScaledDotProductAttentionDecode | 56024 |
| InterleavedToSharded | 1694 |
| NLPConcatHeadsDecode | 1955 |
| Matmul | 24505 |
| ReduceScatter | 24753 |
| Binary | 1517 |
| AllGather | 22055 |
| LayerNorm | 13423 |
| Matmul | 39711 |
| Matmul | 39791 |
| Binary | 15800 |
| Matmul | 68424 |
| ReduceScatter | 24764 |
| Reshard | 3212 |
| Binary | 1496 |
| Total / Layer without flash decode | 387584 |

**Embedding & Lm Head**

| OP CODE | DEVICE FW Duration [ns] |
| --- | --- |
| Embeddings | 5622 |
| Embeddings | 5767 |
| Transpose | 13328 |
| Transpose | 13585 |
| Slice | 2088 |
| Slice | 2193 |
| InterleavedToSharded | 1795 |
| InterleavedToSharded | 1703 |
| AllGather | 24259 |
| LayerNorm | 12906 |
| Matmul | 546508 |
| ShardedToInterleaved | 8775 |
| Total LM Head | 638529 |

*Table 7.* Llama 3.1 8B latency breakdown Tensor Parallel on 8 x N150 chip.

**Per Layer**

| OP CODE | DEVICE FW Duration [ns] |
| --- | --- |
| AllGather | 34069 |
| LayerNorm | 12507 |
| Matmul | 35745 |
| ShardedToInterleaved | 1620 |
| NLPCreateHeadsDecode | 3976 |
| RotaryEmbeddingLlama | 4839 |
| RotaryEmbeddingLlama | 4837 |
| PagedUpdateCache | 5007 |
| PagedUpdateCache | 5078 |
| ScaledDotProductAttentionDecode | 50096 |
| InterleavedToSharded | 1708 |
| NLPConcatHeadsDecode | 1774 |
| Reshard | 2274 |
| AllGatherMatmul | 88825 |
| Reshard | 2847 |
| Binary | 1464 |
| AllGather | 54386 |
| LayerNorm | 15771 |
| Matmul | 22705 |
| Matmul | 22899 |
| Binary | 15804 |
| Matmul | 38486 |
| ReduceScatter | 59572 |
| Reshard | 2817 |
| Binary | 1441 |
| Total / Layer without flash decode | 440451 |

**Embedding & Lm Head**

| OP CODE | DEVICE FW Duration [ns] |
| --- | --- |
| Embeddings | 5641 |
| Embeddings | 5607 |
| Transpose | 13717 |
| Transpose | 13680 |
| Slice | 2095 |
| Slice | 2092 |
| InterleavedToSharded | 1707 |
| InterleavedToSharded | 1726 |
| AllGather | 43463 |
| LayerNorm | 12909 |
| Matmul | 276107 |
| ShardedToInterleaved | 5716 |
| Total LM Head | 384460 |

# G. [Section 2]{.blue} Needle in a Haystack Experiment Details

The tiny needle in a haystack experiment is served as a motivational experiment rather than a rigorous benchmark. It is conducted on a custom dataset created with the following method:

- First, we create a large context using the text of the paper "Compute Substrate for Software 2.0" (Vasiljevic et al., 2021) and the first 4 chapters of the book "A Tale of the Two Cities" by Charles Dickens. This makes the total context length around 16K.

- Then, we generate 7 random MD5 hashes and insert at random places in the text.

- To ensure the result is not biased towards hash strings, we make up 3 random facts and insert at random places in the text.

We then prompt the model with the following questions after the context:

- Start with text "Answer the following questions if you can. If you don't know the answer, answer I don't know."

- Then, we put questions about fact 1, 2, and 3.

- In the end, we ask the model to repeat hashes 1-7.

