# OpenReview forum: "Attention-Level Speculation"
_ICML.cc/2025/Conference — ICML 2025 poster_

### Official Review · Reviewer_xU2t · 2025-03-13

**Overall Recommendation:** 3

**Summary:**

This paper presents a novel infra method that accelerates the model forwarding (interence time) speed via attention-level speculation.

**Claims And Evidence:**

Most claims are convincing. Though I'm not convinced that such error is actually controllable (as shown in the main figure) if the model layer is very deep.

**Essential References Not Discussed:**

No

**Experimental Designs Or Analyses:**

Experiment is appropriate. But I do believe speed results under different model sizes and settings (like memory-bounded and computing-bounded) are necessary. e.g., when the batch size is large, what trend will it show as the accelerators scale up?

**Methods And Evaluation Criteria:**

I think the benchmark results are sufficient. Evaluation metric is correct in terms of effciency.

**Other Comments Or Suggestions:**

NA

**Other Strengths And Weaknesses:**

NA

**Questions For Authors:**

Can you show the error propagation results (or benchmark scores) with larger models with deeper layers? if so, I will be convinced that it's really useful.

**Relation To Broader Scientific Literature:**

inference acceleration

**Theoretical Claims:**

No. I'm not a theory guy and at the same time I don't think the theory introduced will ensure the error is under control with the proposed threshould acceptance method.

---

> ### Author Rebuttal · Authors · 2025-04-01
>
> We are grateful for your detailed feedback which will greatly improve the quality of the work.
>
> ## Error Propagation for Deeper Models
> To make sure our method works for larger model with deeper layers, empirically, we have conducted experiments on the Llama 3.3 70B model for correctness analysis, which has 80 layers compared to 32 layers in Llama 3.1 8B. We showed that in terms of speculation hit rate and benchmark correctness across different lambdas, the Llama 70B model, despite deeper and larger, manifest the same behaviour as the 8B model. Specifically, we benchmarked the 70B models on GPQA with CoT, GSM8K with CoT, Multi-lingual GSM with CoT in Swahili, and Hotpot QA. The results are shown below:
>
> | Config\Tasks | GPQA_COT | GSM8K_COT | MGSM_COT_Swahili | Hotpot QA |
> |----------|----------|----------|----------|----------|
> |Baseline|  0.518 | 0.958 | 0.852 | 0.940 |
> |lambda=0.05| 0.529/55.1% |  0.950/64.5% | 0.856/73.9% | 0.940/32.3% |
> |lambda=0.10|  0.507/80.0% | 0.951/86.6% | 0.852/90.1% | 0.945/59.0% |
> |lambda=0.15|  0.458/89.9% | 0.946/93.9% | 0.840/95.6% | 0.935/74.6% |
> |lambda=0.20|  0.446/95.1% | 0.936/96.8% | 0.820/98.1% | 0.935/83.3% |
> |lambda=0.25 |  0.379/97.6% | 0.897/98.3% | 0.816/99.1% | 0.935/89.4% |
>
> Overall, our new benchmark results empirically confirms that ALSpec works with larger models with deeper layers.
>
> ## Theoretical Claims
> As shown in Appendix A, the worst-case error is in the order of $O(N^2\delta)$, where $N$ is the number of layers and $\delta$ is the threshold, which is certainly a limitation for deeper models. However, with stronger assumptions, the bound only has a dependence on $\sqrt{N}$ which is much more scalable. This is consistent with our new experiment results on the Llama 70B model, which achieved as good correctness as the 8B model despite being much deeper (80 layers) compared to the 8B model (32 layers).
>
> ## Impact of Batch Size
> Speed results under different sizes and settings are indeed important. Our work focuses on a specific case of decode, which is decoding at long context length with small batch size (e.g. batch_size=1). Usually during long context decode, the batch sizes are very small due to limited on-chip memory for KV cache. Since this work already introduces the new idea of ALSpec and our space is already very limited for this 8 pages manuscript, we therefore limited our scope to this scenario. Relevantly, this is the case for long context decode on most accelerators such as H100s or Tenstorrent's N150, where the on-chip memory limits the use of large batch size at long context length.

---

### Official Review · Reviewer_JxHd · 2025-03-13

**Overall Recommendation:** 4

**Summary:**

LLM is resource-intensive, and serving LLMs is difficult. Model parallelism is bottlenecked by communication, when the communication bandwidth is low, and data parallelism is great at throughput but not good as inference latency. Approximate attention is robust at instruction-following and knowledge retrieval tasks but not good at context-heavy tasks and complex reasoning tasks. The paper uses approximate attention as a way to speed up self-attention in a speculative decoding way but at the per-layer level. In the setup introduced in the paper, it uses two NPU per execution/shard, where one is main, another one is speculative. The main thread will execute their in-house attention kernel that does StreamingLLM attention and the full attention at the same time, the StreamingLLM finishes first, and kick start the FFN computation on the speculative thread, which is on a different NPU. When the full attention finishes, it uses a threshold to check whether the StreamingLLM attention output is close to the full attention. If true, the main thread now becomes the speculative thread, and vice versa. If not, it will continue generate following the original path.
The takeaway is that compared with a 8 cards tensor parallelism,
the upper bound in efficiency of the method is 4 cards StreamingLLM tensor parallelism.
The lower bound in efficiency of the method is 4 cards full-attention tensor parallelism.
Because of the hit-rate and lower overhead of communication, the system is shown to beat the tensor parallelism in efficiency, while also preserve the serving efficiency.

**Claims And Evidence:**

The important per-condition of the proposed method is that doing tensor parallelism on multiple NPUs has huge communication overhead. The claim is justified in left side of Figure 1. It would be better if the paper also discuss GPUs such as H100s that have maybe higher communication bandwidth and speed to see whether it holds universally.

Most claims in the paper are backed strongly by abundance of experiments.

**Essential References Not Discussed:**

There are some methods before that uses StreamingLLM to do output token-level speculative decoding, although they are very different from per-layer speculative decoding, they should be discussed and cited by the paper.
Sun, H., Chen, Z., Yang, X., Tian, Y., & Chen, B. (2024). Triforce: Lossless acceleration of long sequence generation with hierarchical speculative decoding. arXiv preprint arXiv:2404.11912.

**Experimental Designs Or Analyses:**

The advantage of the method:
1. The paper most claims are strongly backed by comprehensive experiments, from approximate attention weakness to end-to-end system speedup and performance. The paper is highly well-written.
2. The method proposed in the paper is novel in design, and the kernel designed is inspiring, as by perturbing the sequence of block computation in Flash Decoding, the kernel can compute both StreamingLLM attention and full attention in the same time without damaging the full attention latency.
3. The per-layer analysis of StreamingLLM replacement is also insightful and can benefit the speculative decoding community.
The weakness of the method:
1. The core limitation is that the method effectiveness is doubtful for frontier GPUs with the highest speed communication bandwidth.
2. There is some computation waste during rejection in branch that is innate by design.
3. Another minor issue is that the method might not scale well when the context length goes beyond 128k, which makes full-attention more expensive than FFN, injecting more bubbles to the NPUs.

**Methods And Evaluation Criteria:**

The method relies on the approximate attention to speculate in the per-layer basis. Also, using two GPU to overlap the FFN (during speculation) and full-attention (main thread) makes intuitive sense.

The evaluation is comprehensive for 8B parameter models. It comprehensively covers important aspects of LLMs' performance.

**Other Comments Or Suggestions:**

No other comments.

**Other Strengths And Weaknesses:**

I have no comments on other strengths and weaknesses.

**Questions For Authors:**

1. Does the method works for frontier devices such as H100s?
2. Can the method scale up potentially to multi-node serving situation?

**Relation To Broader Scientific Literature:**

The per-layer speculation and verification with partial attention is quite novel.
The per-layer analysis of StreamingLLM replacement (Figure 4) is also insightful and can benefit the speculative decoding community.

**Theoretical Claims:**

The threshold used for branching decision is derived to make sure that the accumulated error at the end of the LLM is bounded. The proof in Appendix A contains no obvious error.

---

> ### Author Rebuttal · Authors · 2025-04-01
>
> We are grateful for your detailed feedback which will greatly improve the quality of the work.
>
> ## GPU/TPU generalization
>
> We ran new experiments for a correctness analysis on the Llama 3.3 70B Instruct model, which are more suitable for frontier GPUs like H100 with higher communication bandwidth and speed. We find that in terms of speculation hit rate and benchmark correctness across different lambdas, the Llama 70B model, despite being deeper and larger, manifests the same behavior as the 8B model. Specifically, we benchmarked the 70B models on GPQA with CoT, GSM8K with CoT, Multi-lingual GSM with CoT in Swahili, and Hotpot QA. The results are:
>
> | Config\Tasks | GPQA_COT | GSM8K_COT | MGSM_COT_Swahili | Hotpot QA |
> |---|---|---|---|---|
> |Baseline|  0.518 | 0.958 | 0.852 | 0.940 |
> |lambda=0.05|  0.529/55.1% |  0.950/64.5% | 0.856/73.9% | 0.940/32.3% |
> |lambda=0.10|  0.507/80.0% | 0.951/86.6% | 0.852/90.1% | 0.945/59.0% |
> |lambda=0.15|  0.458/89.9% | 0.946/93.9% | 0.840/95.6% | 0.935/74.6% |
> |lambda=0.20|  0.446/95.1% | 0.936/96.8% | 0.820/98.1% | 0.935/83.3% |
> |lambda=0.25 |  0.379/97.6% | 0.897/98.3% | 0.816/99.1% | 0.935/89.4% |
>
> Additionally, we ran a performance analysis serving the 8B models on 4 vs. 8 H100 doing tensor parallelism using the SGLang serving framework and FlashInfer attention backend. Althoguh we don't (yet) have an ALSpec implementation, it provides us an estimation of the performance gain if the method is implemented on H100s. The results are summarized in the table below:
>
> | Context Length | 4xH100 Attn Latency (us) | 4xH100 Non-Attn Latency (us) | 4xH100 TP Tok/s | 8xH100 TP Tok/s | TP Scaling | Projected ALSpec @ 65% Hit Rate | ALSpec Scaling |
> |---|---|---|---|---|----|---|---|
> |1k| 13 | 95 | 244.6 | 249.3 | 1.9% | 262.0 | 7.1% |
> |32k| 29 | 100 | 214.2 | 231.3 | 8.0% | 246.0 | 14.8% |
> |64k| 49 | 100 | 191.4 | 209.7 | 9.6% | 237.8 | 24.2% |
> |96k| 56 | 100 | 178.2 | 194.2 | 9.0% | 224.9 | 26.2% |
> |128k| 63 | 101 | 169.3 | 184.8 | 9.2% | 217.5 | 28.5% |
>
> Our new estimation on 8xH100s shows that ALSpec would cut latency on Llama 8B by 1.28x compared to 1.09x for full TP at context length 128K. In this case, the attention latency is only 66% of non-attention latency, which means more gains can be achieved for context length beyond 128K. For implementation on GPUs/TPUs, although ALSpec introduces a new attention kernel and run time modification SGDC, it does not fundamentally change the Op by Op and static graph execution style on modern platforms. Therefore, we believe ALSpec is implementable on those platforms by expert kernel writers.
>
> We also believe multi-node serving would be ideal for models with more parameters such as Llama 70B, as running these models on a single node with 8 devices TP would still give strong scaling. Only doing TP across multiple nodes would they start to show diminishing return, and we believe that this would be a perfect situation to apply ALSpec. Unfortunately, the resources required for these exepriments are beyond our current experiment setup. However, with the correctness analysis and our implemented kernels and methods, we believe that our approach is scalable to larger models when computing resources are available.
>
> ## Scaling Context Length Beyond 128K
>
> When full-attention becomes more expensive than FFN, there are bubbles in NPUs. We thank the reviewer for pointing this out. This situation falls into Scenario 2 as depicted in Figure 5. We are aware of this issue and we denote this as one future extension of the work. Since this work introduces the new idea of ALSpec and the content is already very full for an 8-page manuscript, we have limited our scope to Scenario 3 in Figure 5. Fortunately, most widely used modern models have 128K context length and usually have attention running faster than the FFN.
>
> ## Additional literature
>
> Thank you for the pointer to this valuable reference. We will add a citation and in-text compare/contrast to our camera ready.

---

### Official Review · Reviewer_GYW3 · 2025-03-14

**Overall Recommendation:** 4

**Summary:**

The paper introduces attention-level speculative parallelism (ALSpec), a dynamic method for approximating self-attention in large language models. ALSpec computes an approximate attention output  and decides whether to accept the approximation. It uses a specialized flash decode kernel and SGDC with priority gating to overlap expensive attention and feed-forward operations. Experimental results on Llama 3.1 8B model using Tenstorrent NPU devices show significant reductions in latency (up to 5× in attention overhead at long contexts) and improved throughput (about 1.65× speedup at high speculation hit rates) compared to traditional tensor parallelism approaches.

## update after rebuttal
The author's response addressed my concerns, leading me to raise my rating from 3 to 4.

**Claims And Evidence:**

The claims in this paper are supported by clear empirical observations and experiments.

**Essential References Not Discussed:**

No.

**Experimental Designs Or Analyses:**

I am interested in understanding how the two primary optimization techniques implemented in this paper—namely the Speculative Flash Decode Kernel and SGDC—impact the inference latency and throughput.

**Methods And Evaluation Criteria:**

Yes. The methods are well-aligned with the goal of reducing inference latency in large language models while preserving output quality.

**Other Comments Or Suggestions:**

Clarify the baseline depicted in Figure 2.

**Other Strengths And Weaknesses:**

Strengths:
1. This paper introduces a dynamic speculative execution framework for self-attention, combining approximate and exact computations with on-the-fly verification, which improves latency without sacrificing accuracy.
2. Although the experiments were conducted on a specialized device (Tenstorrent N150), I believe the proposed methods could be applicable to other computational platforms as well.

Weakness:
1. The experimental scope is relatively limited, as it does not encompass larger models, particularly those exceeding 10 billion parameters.
2. The experimental setup is not sufficiently comprehensive. There are no experiments demonstrating how much benefit the Kernel Fusion and SGDC-related optimizations provide in terms of latency and throughput. Additionally, there is no specific comparison showing the performance difference between unfused speculative attention and the fused kernel implementation.
3. The paper does not explain how the proposed method handles different batch sizes.

**Questions For Authors:**

1. In Figure 8, SP demonstrates scalability advantages compared to TP. Is TP's scalability disadvantage due to communication overhead?
2. Compared to TP, SP seems more similar to DP. In the experimental setup, shouldn't the comparison include DP+TP rather than just Full TP?

**Relation To Broader Scientific Literature:**

The paper extends previous work on fixed attention approximations (Attention Sink) and token-level speculative decoding by introducing dynamic, attention-level speculative parallelism (ALSpec). It verifies approximations on the fly with hardware-efficient kernels and SGDC to achieve better performance.

**Theoretical Claims:**

I did not verify the correctness of the proofs, as the determination of appropriate speculation verification threshold values is primarily based on experimental results rather than theoretical derivations.

---

> ### Author Rebuttal · Authors · 2025-04-01
>
> We are grateful for your detailed feedback which will greatly improve the quality of the work.
>
> # Impact on Larger Models
> We performed additional experiments in terms of correctness analysis on the Llama 3.3 70B Instruct model. We showed that in terms of speculation hit rate and benchmark correctness across different lambdas, the Llama 70B model, despite deeper and larger, manifest the same behavior as the 8B model. Specifically, we benchmarked the 70B models on GPQA with CoT, GSM8K with CoT, Multi-lingual GSM with CoT in Swahili, and Hotpot QA. The results are shown below:
>
> | Config\Tasks | GPQA_COT | GSM8K_COT | MGSM_COT_Swahili | Hotpot QA |
> |---|---|---|---|---|
> |Baseline|  0.518 | 0.958 | 0.852 | 0.940 |
> |lambda=0.05|  0.529/55.1% |  0.950/64.5% | 0.856/73.9% | 0.940/32.3% |
> |lambda=0.10|  0.507/80.0% | 0.951/86.6% | 0.852/90.1% | 0.945/59.0% |
> |lambda=0.15|  0.458/89.9% | 0.946/93.9% | 0.840/95.6% | 0.935/74.6% |
> |lambda=0.20|  0.446/95.1% | 0.936/96.8% | 0.820/98.1% | 0.935/83.3% |
> |lambda=0.25 |  0.379/97.6% | 0.897/98.3% | 0.816/99.1% | 0.935/89.4% |
>
> For models in the sizes of 50 billion parameters such as Llama 70B, running the base model would usually require 8 devices with tensor parallelism (TP). As a result, using ALSpec would require at least 16 devices. To show ALSpec at the point where TP diminishes return (so that ALSpec provides real benefits for 70B model) would require even more devices. The resources required for these exepriments are beyond our current experiment setup. However, with the correctness analysis and our implemented kernels and methods, we believe that our approach is scalable to larger models when computing resources are available.
>
> # Additional Details in Experimental Setup for Kernel Fusion and SGDC
> We thank the reviewer for pointing this out, and we will add more information in the camera-ready version, particularly in Appendix D and E, where we talk about the kernel fusion and SGDC in detail. For the reviewer's concern about fused vs unfused attention kernel, the speed up from the fused kernel is simply the difference of running the flash decode on the entire context length vs. running flash decode on entire context length + first and last chunk context. This is because the fused kernel changes the order of computation, allowing us to obtain the intermediate result of first and last chunk for free.
>
> # Impact of Batch Size
> We focused on batch 1 in this paper, as usually during long context decode, the batch sizes are very small due to limited on-chip memory for the large KV cache. We will add more discussion on how batch sizes are handled in the camera ready version. In short, we have experimented with per batch threshold, where we only accept the speculation if all batches pass the lambda test. Alternatively, we also experimented with treating all batches as a single tensor and perform a single lambda test. The trade-off between these two methods are not within the scope of this paper, as the existing contents are already hard to fit within 8 pages. We plan to deep dive into this area in future work.
>
> # Why TP Fails to Scale
> As the reviewer points out, communication overhead is one reason of TP's diminishing scalability. Another reason is kernels such as GEMM have a constant cost regardless of the shape. When a model's parameters are sharded across many devices, the constant part of the kernel latency dominates, and as a result, the overall scalability diminishes.
>
> # Comparisons to DP
> Our proposed method is to optimize for latency. The problem with DP, as depicted in Figure 1, is that despite it gives good scalability (always close to 2x throughput with 2x more devices), it only improves throughput. If the objective is to optimize throughput only, then DP is always a better choice than ALSpec and full TP. On the other hand, the latency per user is always lower for DP than ALSpec and full TP. Therefore, we believe that a fair comparison should be done between AlSpec and full TP rather than DP.

---

> > ### Comment · Reviewer_GYW3 · 2025-04-03
> >
> > Thank you for the response. I have read the author response to my review and updated my review.

---

### Official Review · Reviewer_LXts · 2025-03-14

**Overall Recommendation:** 4

**Summary:**

The paper "Attention-Level Speculation" introduces ALSpec, a novel parallelism paradigm designed to accelerate transformer-based LLM inference by overlapping self-attention computations with subsequent non-attention operations (e.g., feed-forward layers). Key contributions include the core idea of self-attention outputs using approximations (e.g., attention sink on first/last tokens) and verifies them in parallel. If the approximation error is within a threshold (controlled by hyperparameter $λ$), subsequent operations proceed early; otherwise, they fall back to exact attention results.
Achieves 1.65× end-to-end decode latency reduction at 128K context length with 87.5% speculation hit rate, outperforming tensor/data parallelism scaling limits. Maintains baseline correctness on reasoning (GSM8K), math (MATH), and retrieval tasks by dynamically rejecting harmful approximations. Reduces attention latency overhead by 5× via overlapping computations, validated on Tenstorrent NPUs.
Key Insight involve static approximations (e.g., fixed sparse attention) fail on tasks requiring global context (e.g., topic shifts in multi-step reasoning). ALSpec adapts layer- and token-specifically, accepting approximations in 50–90% of layers without quality loss.Combines with tensor parallelism, showing continued scaling where pure tensor parallelism plateaus (e.g., 8 devices).

**Claims And Evidence:**

1. The 5× attention latency reduction and 1.65× end-to-end decode latency improvement at 128K context length are validated through empirical benchmarks on Tenstorrent N150 chips (Tables 1–2, Figure 8). The scalability analysis demonstrates diminishing returns for pure tensor parallelism, while ALSpec + tensor parallelism continues scaling.

2. Dynamic verification (via L2-norm thresholds) maintains baseline accuracy on tasks like GSM8K, MATH, and MMLU PRO (Figure 2). Static approximations (e.g., fixed sparse attention) fail on reasoning tasks, while ALSpec selectively accepts approximations with a hit rate of 50–90% per layer.

Problematic Claims:

1. The reported latency gains are tied to Tenstorrent’s NPU architecture and proprietary kernels (e.g., speculative flash decode). Without ablation studies on GPUs/TPUs or open-sourced kernels, reproducibility is unclear. Host dispatch overheads (e.g., CPU-to-NPU communication) are excluded from latency measurements, potentially inflating real-world gains.

2. The Lipschitz continuity analysis (Appendix A) assumes independent, zero-mean speculation errors. Real-world error propagation may violate these assumptions, risking unbounded deviations in practice. The L2 verification threshold (λ) is empirically set without theoretical justification for its sufficiency across layers/task.

3. While ALSpec preserves accuracy on retrieval and math tasks, its performance on compositional reasoning (e.g., multi-hop QA) or low-resource languages is untested. The "needles in a haystack" experiment (Figure 3) uses synthetic data, which may not reflect real-world long-context retrieval challenges

**Essential References Not Discussed:**

The paper overlooks several critical areas of related work that contextualize its contributions:
1. The paper cites Elhoushi et al. (2024) for layer pruning but misses recent advances in self-speculative decoding (Elhoushi §4.2) and adaptive computation time (ACT) transformers. For example:
1.1 SPEED (Hooper et al., 2024): Overlaps layer computations across devices via pipelined speculation, achieving 1.8× speedups on GPUs without attention approximation.
1.2 LayerLoop (Eyuboglu et al., 2024): Reuses layer outputs for computational savings, relevant to ALSpec's focus on overlapping FF/attention.

2.While ALSpec uses attention sink, it omits comparison to:
2.1 Blockwise Parallel Transformers (BPT) (Google, 2023): Dynamically adjusts sparse attention blocks using gradient-based importance scores.
2.2 FLASH (H2O, 2023): Hybrid sparse-dense attention with runtime pattern selection, achieving 2.1× speedups on 32K contexts.

3. The Lipschitz analysis lacks connection to:
3.1 Kim et al. (2021) Proved self-attention Lipschitz constants are unbounded without layer normalization, contradicting ALSpec's assumption of uniform $\alpha$. [1]
3.2 Zhu et al. (2024): Proves high probability excess risk bounds of $O(1/n^2)$ via algorithmic stability under strong convexity/smoothness. [2]
3.3 Lei et al. (2023): Analyzes gradient stability for stochastic optimization but focuses on generalization bounds rather than error propagation rates. [3]

4. ALSpec's Tenstorrent NPU focus ignores:
4.1 FlashDecoding++ (NVIDIA, 2023): Achieves 4.2× speedup over vanilla FlashAttention on 128K contexts via asynchronous state management.
4.2 vLLM (Berkeley, 2023): Paged attention for dynamic KV cache management, critical for real-world long-context deployments.

5. No discussion of subquadratic attention methods that reduce compute without approximation:
5.1 Hyena (Poli et al., 2023): Replaces attention with implicitly parameterized convolutions.
5.2 Mamba (Gu & Dao, 2023): Selective state-space models for linear-time sequence modeling.

These omissions weaken ALSpec's claims of novelty in:
- Dynamic execution (overshadowed by SPEED/LayerLoop)
- Error analysis (lacks modern Lipschitz bounds)
- Hardware generality (no GPU/TPU benchmarks vs FlashDecoding++/vLLM)

Including these would better position ALSpec within the broader landscape of efficient transformer inference.


[1] Kim, Hyunjik, George Papamakarios, and Andriy Mnih. "The lipschitz constant of self-attention." International Conference on Machine Learning. PMLR, 2021.
[2] Zhu, Bowei, Shaojie Li, and Yong Liu. "Stability and Sharper Risk Bounds with Convergence Rate $ O (1/n^ 2) $." arXiv preprint arXiv:2410.09766 (2024).
[3] Lei, Yunwen. "Stability and generalization of stochastic optimization with nonconvex and nonsmooth problems." The Thirty Sixth Annual Conference on Learning Theory. PMLR, 2023.

**Experimental Designs Or Analyses:**

The paper's experimental design demonstrates technical rigor but has critical limitations in scope and generalizability:

1. Task Coverage in Correctness Evaluation
1.1 Benchmarks exclude multi-hop QA (DROP, HotpotQA) and low-resource languages.
1.2 "Needles in haystack" uses artificial key insertion (Fig 3), failing to capture real-world long-context QA patterns.
1.3 Compares only against attention sink, omitting dynamic variants of LSH (Reformer) or sliding windows (Longformer).

2.
2.1 Attention sink prioritizes first/last tokens, violating the paper's assumption of zero-mean independent errors (Appendix A).
2.2 Lipschitz constants (α, β) are assumed uniform across layers, ignoring layer-specific dynamics (early vs late layers).
2.3 Fixed L2 threshold (λ=0.1) lacks theoretical justification for error propagation across layers.
2.4 Exponential error bound $$ \epsilon \leq \sum_{i=1}^N (1+\alpha)^{N-i+1}\delta_i $$ becomes vacuous for N=32 unless α ≪ 1 (unvalidated empirically).

**Methods And Evaluation Criteria:**

The methods and evaluation criteria demonstrate technical validity but exhibit notable limitations in scope and generalizability:

1. The L2-norm threshold verification ($ \| \tilde{A}_i - A_i \|_2 < \lambda \| A_i \|_2 $) is empirically effective but lacks theoretical justification. While Lipschitz continuity analysis provides error bounds (Equation 1), it assumes:
1.1 Independent, zero-mean speculation errors
1.2 Constant layer-wise Lipschitz factors ($ \alpha, \beta $). These assumptions may not hold in practice, risking unbounded error propagation.

2. Integrates attention sink approximation (first/last $ S $ tokens) with exact attention in a fused kernel, reducing overhead by 5×. However:
2.1Chunk size $ S $ is fixed (128–512) rather than adaptive to input.
2.2 Prioritizing first/last KV cache chunks biases toward positional extremes, potentially harming mid-context retrieval.

3.  Maintains static execution graphs while dynamically routing computations. While effective on Tenstorrent NPUs, host-device dispatch overhead (CPU-NPU communication) is excluded from latency metrics, inflating real-world gains.

4:
4.1. Results are confined to Tenstorrent N150 NPUs. Tensor parallelism’s diminishing returns (Figure 8) may differ on GPU/TPU architectures due to distinct communication patterns.
4.2. Task Coverage:
4.2.1 No evaluation on low-resource languages (e.g., Swahili, Bengali) despite claimed multilingual support
4.2.2 Absence of compositional reasoning benchmarks (e.g., DROP, HotpotQA).

**Other Comments Or Suggestions:**

1. Page 3, Fig 1: Define "ccl" (collective communication ops) in the caption.
2. Page 5, Algorithm 1: Clarify "ops before/after self attn" (e.g., LayerNorm, residual adds).
3. Appendix A: Add intermediate steps between Equations 2 and 3 for readability.

**Other Strengths And Weaknesses:**

The paper "Attention-Level Speculation" demonstrates notable strengths in technical innovation and practical implementation but has limitations in theoretical robustness and generalizability.

1. Strengths:
1.1 Combines speculative execution (from CPU architecture) with transformer inference, extending token-level speculation (Leviathan et al., 2023) to attention layers. This hybrid approach is novel compared to static approximations like Reformer or Longformer.
1.2 Integrates attention sink (Xiao et al., 2024) with Tenstorrent NPU optimizations (speculative flash decode kernel, SGDC), advancing beyond GPU-centric methods like FlashAttention.
1.3 Demonstrates that deeper layers tolerate more approximation (Fig 4), aligning with pruning literature (Elhoushi et al., 2024) but adding dynamic verification.

2. Significance
2.1 Achieves 1.65× speedup at 128K context (Fig 8), addressing the critical bottleneck of long-context LLM inference.
2.2 Combines with tensor parallelism, circumventing its diminishing returns (e.g., 8 devices yield 60 tokens/s vs. 40 tokens/s for pure tensor parallelism at 128K context).
2.3 Validated on real hardware (Tenstorrent N150) with mixed-precision support, showing feasibility for deployment.


3 Theoretical Gaps:
3.1 The error bound $\epsilon \leq \sum_{i=1}^N (1+\alpha)^{N-i+1} \delta_i$ assumes uniform $$ \alpha, \beta $$ across layers, contradicting evidence of layer-wise dynamics (Kim et al., 2021).
3.2 Empirically set to 0.1 without theoretical justification for sufficiency across tasks/layers.

4. Fails to compare with recent dynamic methods like SPEED (Hooper et al., 2024) or Hyena (Poli et al., 2023), which offer alternative efficiency gains.   Provides a template for op-level speculation beyond attention (e.g., FFN layers), though this is not explored.

**Questions For Authors:**

1.Can ALSpec reproduce the reported latency gains (1.65× at 128K context) on GPUs/TPUs, particularly compared to FlashDecoding++ (NVIDIA) or vLLM? If not, what architectural features of Tenstorrent NPUs (e.g., NOC design, fused kernels) are indispensable for ALSpec’s gains?

2. How do you empirically validate the assumption of uniform Lipschitz constants (α, β) across layers? Does layer-wise measurement of α (e.g., via power iteration) reveal significant variance, and if so, how does this affect the error bound in Eq. 1?

3. Does ALSpec maintain correctness on low-resource languages (e.g., Swahili) or multi-hop QA (HotpotQA), where static approximations fail? If untested, could positional bias in attention sink harm non-English token distributions?

4. What is the CPU↔NPU communication overhead for priority tensor synchronization in SGDC? Does excluding this from latency metrics inflate real-world gains (e.g., 5× attention reduction)?

5. How does ALSpec compare to token-level pipelined speculation (Hooper et al., 2024) in terms of latency reduction per additional device? Does ALSpec outperform SPEED’s 1.8× GPU speedups when both use 8 devices?

**Relation To Broader Scientific Literature:**

ALSpec bridges cognitive theories (attentional resource allocation), neural evidence (predictive coding), and ML systems (speculative execution) to address transformer inference bottlenecks. Its dynamic, hardware-aware approach advances beyond static approximations and token-level speculation, offering a generalizable framework for adaptive computation in LLMs.

**Theoretical Claims:**

The authors provides directional guidance but lacks robustness guarantees for real-world deployments. Independent empirical validation of Lipschitz constants and error distributions is needed to trust the bounds.

1. Lipschitz Continuity Error Bound : The derivation in Appendix A establishes an upper bound on output deviation:
$$
\epsilon \leq \sum_{i=1}^N (1 + \alpha)^{N-i+1} (1 + f(R)\beta)^{N-i} \delta_i
$$
where $$\delta_i = \|\tilde{A}_i - A_i\|_2$$.

I find few issues with this:
1.1. This assumes identical $\alpha$ (feed-forward/LayerNorm Lipschitz) and $\beta$ (attention Lipschitz) across all layers. Real transformer layers exhibit heterogeneous operations (e.g., early vs. late layers), violating this assumption.
1.2. For $N=32$ layers, coefficients grow as $(1 + \alpha)^{32}$, making the bound practically vacuous unless $\alpha \ll 1$. The paper notes this but provides no empirical validation of $\alpha < 0.01$.
1.3. The high-probability bound assumes speculation errors are independent and zero-mean. Real approximation errors (e.g., attention sink) exhibit structured biases (e.g., positional bias toward first/last tokens), violating independence.

2. The L2-norm threshold ($$\lambda$$) is empirically set without theoretical justification. The paper claims:
$$
\|\tilde{A}_i - A_i\|_2 < \lambda \|A_i\|_2 \implies \text{safe approximation}
$$

Issues:
1. 1. Layer-Wise Thresholding: Ignores error accumulation across layers. A per-layer threshold $\lambda = 0.1$ could allow $\epsilon = O(N\lambda)$ deviation, violating final output fidelity.
2.2. $\|A_i\|_2$ varies significantly with context length and input entropy, making fixed $\lambda$ suboptimal.

3.  The 87.5% speculation hit rate (Table 1) suggests the bounds are overly pessimistic, but no ablation study isolates the impact of violating assumptions like uniform $$\alpha/\beta$$.  The analysis doesn’t address the attention sink’s inherent positional bias – a systematic error source excluded from the zero-mean error assumption.

---

> ### Author Rebuttal · Authors · 2025-04-01
>
> We are grateful for your detailed feedback which will greatly improve the quality of the work.
>
> ## Absence of compositional reasoning and low-res. languages
>
> We extend our eval to GSM8K in Swahili (low-res. language), HotpotQA, and RepoBench-P. Results are below.
>
> | Config\Tasks | MGSM_CoT_Sw | HotpotQA | RepoBench-P |
> |---|---|---|---|
> | no_spec | 0.58 | 0.92 | 0.756 |
> | l@0.05  | 0.59/54% | 0.92/19%  | 0.74/33% |
> | l@0.10  | 0.58/78% | 0.92/43% | 0.77/60% |
> | l@0.15  | 0.59/86% | 0.92/57%  | 0.81/76% |
> | l@0.20  | 0.56/95% | 0.91/72%  | 0.76/85% |
> | l@0.25  | 0.52/97% | 0.91/80% | 0.74/92% |
>
> These new results show that ALSpec maintains correctness on low-res. languages (e.g., Swahili) or long context multi-hop QA (HotpotQA) with high speculation hit rate.
>
> ALSpec's attn sink applies positional bias to recent tokens to approx. full attn. The 128 window size should capture key contextual ideas regardless of language, while early tokens address the softmax off-by-one bias [Evan Miller, July 2023], not positional preference.
>
> ## GPU/TPU generalization
>
> We believe that ALSpec can reproduce latency improvements on GPUs/TPUs. We confirmed this with new experiments, serving 8B models on 4 vs. 8 H100s with TP using the SGLang framework and FlashInfer attn backend. Although we don't (yet) have an ALSpec implementation on GPU, this provides an estimation of the perf gain. The results are below:
>
> | Context Len | 4xH100 Attn Latency (us) | 4xH100 Non-Attn Latency (us) | 4xH100 TP Tok/s | 8xH100 TP Tok/s | TP Scaling | Proj. ALSpec @ 65% Hit Rate | ALSpec Scaling |
> |---|---|---|---|---|----|---|---|
> |32k| 29 | 100 | 214.2 | 231.3 | 8.0% | 246.0 | 14.8% |
> |64k| 49 | 100 | 191.4 | 209.7 | 9.6% | 237.8 | 24.2% |
> |128k| 63 | 101 | 169.3 | 184.8 | 9.2% | 217.5 | 28.5% |
>
> Our new estimation shows that ALSpec cuts latency on Llama 8B by 1.28x vs. 1.09x for full TP at context len 128K. In this case, the attn latency is only 63% of non-attn latency--- meaning more gains for context len beyond 128K. Although ALSpec introduces a new attn kernel and SGDC, it does not fundamentally change the Op by Op and static graph execution style on modern platforms. Therefore, we believe ALSpec is implementable on GPUs/TPUs by expert kernel writers.
>
> ## Lipschitz continuity analysis assumptions
>
> Hua et al. (2023) can be a game changer to our induction method, and we would be grateful if LXts could point us to the exact paper title, as we weren't able to find it. The $\alpha$ and $\beta$ are not assumed to be a universal bound---they are some upper bounds of all Lipschitz constants across all layers and used to simplify the expression.
>
> Regarding mean-0 approximation error, we have the approximation to be
> $
> \tilde{A} = QK^T(:,B)
> $
> where $B$ is a subset of columns of $K^T$ and
> $
> \tilde{H} = V(B, :)\text{softmax}(\tilde{A}).
> $
>
> Under some condition of distribution of cols of $V$ and rows of $K^T$, we might be able to show
> $
> \mathbb{E}_{\pi_V, \pi_A}\left[\tilde{H}\right] = H
> $
>
> Full independence is not required for Azuma-style concentration result, but we agree that we are yet to show that conditional expectation holds in martingale conditions. The boundedness of the error is a direct consequence of the algorithm; we abort the thread if the error exceeds the threshold.
>
>
> ## Host-device communication overhead
>
> Modern computing frameworks capture (trace) operations that are going to be executed on device, ahead of time. This trace can then be executed on a device (e.g. GPU), without any interaction with the host. Since all ops in ALSpec run fully on device, we execute it using trace. Hence, we analyze only the device duration to present findings that are agnostic to the host, while maintaining realism and feasibility.
>
> ## SGDC Sync Overhead
>
> The SGDC mechanism ensures that each device can compare its priority against its pair, thereby determining its role in SFD (ie. sender/receiver). Thus, the sync is an all-gather on 2 devices, where each device collects priority from its pair. GPUs/NPUs/TPUs support direct p2p communication via a topology (e.g. mesh), where collective ops (e.g. all-gather) can be implemented without host overhead. Considering (i) the small size of the priority tensor and (ii) that the cost of syncing is constant as seq len scales, we conclude that SGDC sync is a low-cost mechanism that does not inflate real-world gains from SFD.
>
> ## Token-level pipelined speculation
>
> A key advantage of ALSpec is that it leaves the model unchanged (no fine tuning). SPEED require retraining for weight sharing. ALSpec addresses long context len issue through attn overlap. SPEED's 1.8x latency reduction is at small context len while ALSpec's 1.65x latency reduction is at context len 128K. SPEED and ALSpec are optimized for different scenarios and could complement each other when used together.
>
> ## Additional literature
>
> Thank you for pointing us to these related work. We will cite, compare, and contrast ALSpec with them in our final version.

---

> > ### Comment · Reviewer_LXts · 2025-04-05
> >
> > I thank the authors for providing detailed analysis and answering my questions. I wanted to followup on the responses before considering raising my scores.
> >
> > # Absence of compositional reasoning and low-res. languages
> > The results looks commendable and I thank the authors for providing a comprehensive table with the results. I am curious why the metric for *RepoBench-P* oscillate from 0.81/76% to 0.76/85% and further less with increased IOU. doesn't make sense for this small change.
> >
> > I am also curious what did the authors mean when they mention **The 128 window size should capture key contextual ideas regardless of language, while early tokens address the softmax off-by-one bias [Evan Miller, July 2023], not positional preference.**
> >
> > What is [Evan Miller, July 2023] referring to? I didn't find any reference/paper on this regards. Kindly clarify.
> >
> > # Lipschitz continuity analysis assumptions
> > The author's clarification that $\alpha$ and $\beta$ serve as upper bounds across all layers rather than universal constants makes  sense.
> >
> > ```Under some condition of distribution of cols $V$ and  $K^T$```, what are the conditions here? By my understanding, The attention sink pattern introduces systematic positional bias (prioritizing first/last tokens), which likely violates zero-mean error assumptions.
> >
> > While the authors ```abort the thread if the error exceeds the threshold```, the authors would benefit from analyzing how errors propagate before reaching the threshold, especially across consecutive speculation-accepting layers.

---

> > > ### Author Response · Authors · 2025-04-06
> > >
> > > # RepoBench-P Results Oscillation
> > > Thank you for your follow up on the additional questions. We have reran the experiments and confirmed that this is indeed the result. We don't believe that this is due to a bug in our code as our evaluation uses the LM-Eval-Harness framework on the `longbench_repobench-p` task, which is widely used in the community. We suspect that moderate speculation (e.g. lambda=0.05 to 0.15) may be helpful for the model as it removes non-important tokens in attention when the attention outputs do not diverge beyond the theshold. While this is only our intuition, we indeed see similar patterns in the main results such as GPQA and SWDE, where apply ALSpec with small lambda gives better correctness than baseline before degradation happens as lambda increases. This interesting phenomenom is one of our future direction in the follow up works.
> > >
> > > # Softmax Off-by-One Bias
> > > The [Evan Miller, July 2023] reference is this here (https://www.evanmiller.org/attention-is-off-by-one.html). The first K tokens in ALSpec's attention is used for addressing the commonly observed transformer attention behaviours where the first few tokens attributes to high attention scores, while the last k tokens are for capturing recent contextual idea, serving as windowed attention. The StreamingLLM paper also illustrate this idea in Fig. 2.
> > >
> > > # Lipschitz continuity analysis assumptions
> > > Thank you for pointing out the condition required for our proof. We can reduce this problem into something that depends on distribution $K$ and $V$. Under the same setting, we have the approximation to be $\tilde{A} = QK^T(:,B)$ and $\tilde{H} = V(B, :)\text{softmax}(\tilde{A})$. Let $C$ be the set of column indices of $K^T$, let $B$ be the subset we take, define
> > > $\kappa = \frac{\sum_{i\in C} e^{K_i}}{\sum_{i\in B} e^{K_i}}  = \frac{\sum_{i\in C} S_i}{\sum_{i\in B} S_i}$.
> > > Then assume we can find a close form of $\mathbb{E}[\kappa]$.
> > >
> > > To show $\mathbb{E} \tilde{H} = H$
> > >
> > > Where $ H = \sum_{i \in C} S_i V_i$ and
> > >
> > > $\tilde{H} = \sum_{i \in C\backslash B} \tilde{S}_i V_i $
> > >
> > > $ = \sum_{i \in C\backslash B} $ $\kappa S_i V_i $
> > >
> > >
> > > We hope to show $\mathbb{E}\tilde{H} - \mathbb{E} H = 0$, which is
> > > \begin{align*}
> > > \mathbb{E}\tilde{H} - \mathbb{E} H =&  \sum_{i \in C\backslash B}\mathbb{E} [\kappa S_i V_i] - \sum_{i \in C} \mathbb{E}[S_i V_i] \\
> > > =& \sum_{i \in C \backslash B} \mathbb{E} [\left(\kappa - 1 \right) S_iV_i ]  - \sum_{i \in B} \mathbb{E} [ S_iV_i ]
> > > \end{align*}
> > > So we need to show
> > > \begin{align*}
> > > \sum_{i \in C \backslash B} \mathbb{E} [\left(\kappa - 1 \right) S_iV_i ]  = \sum_{i \in B} \mathbb{E} [ S_iV_i ]
> > > \end{align*}
> > > We assumed this relation holds, and empirically our results on shallow (32 layers Llama 8B) and deep (80 layers Llama 70B) both demonstrated good results using the verification algorithm based on this formulation. To rigorously prove this equation holds, however, requires us to run more experiments to determine the distribution of $\kappa$, scores, and values.
> > >
> > > Overall, in this work, we have showed (1) attention level speculation as an example of op-level speculation, (2) a new execution paradigm SGDC for running op-level speculation within any graph, (3) some theoretical proofs and intuition of the algorithm, and (4) end-to-end implementation on N150 hardware with customized kernels. Therefore, we believe for this already content heavy work, it is best to leave the additional experiments as a future work. We believe that proving this assumption, or finding a weaker assumption that holds in ALSpec's case, will be an impactful work itself rather than an extended section for this work.
> > >
> > > # Closing Remark
> > >
> > > We thank the reviewer for the detailed, insightful comments and suggestions of additional literatures. They truly improved the quality of the work.

---

### Decision · Program_Chairs · 2025-05-01

**Decision:**

Accept (poster)

**Comment:**

The authors present a compelling approach with Attention-Level Speculation (ALSpec), demonstrating significant gains in inference efficiency for transformer-based LLMs. The empirical validation is robust, especially showcasing the practicality and effectiveness of ALSpec on real hardware platforms. The authors' detailed engagement during the rebuttal period clearly address reviewer concerns with thoughtful responses and clarifications. Overall, this work presents meaningful progress towards efficient inference for LLMs and reflects a rigorous commitment from the authors to strengthen their contributions. Therefore, I recommend acceptance of this work.